# Higher resolution pooled genome-wide CRISPR knockout screening in Drosophila cells using integration and anti-CRISPR (IntAC)

Raghuvir Viswanatha[1] ✉, Samuel Entwisle[1], Yanhui Hu [1], Ah-Ram Kim [1], Kelly Reap[1], Matthew Butnaru[1,2], Mujeeb Qadiri[1], Stephanie E. Mohr [1] & Norbert Perrimon [1,2] ✉

CRISPR screens enable systematic, scalable genotype-to-phenotype mapping. We previously developed a CRISPR screening method for *Drosophila melanogaster* and mosquito cell lines using plasmid transfection and site-specific integration to introduce single guide (sgRNA) libraries. The method relies on weak sgRNA promoters to avoid early CRISPR-Cas9 activity causing discrepancies between genome edits and integrated sgRNAs. To address this issue and utilize higher strength sgRNA expression, we introduce a method to co-transfect a plasmid expressing anti-CRISPR protein to suppress early CRISPR-Cas9 activity which we term "IntAC" (integrase with anti-CRISPR). IntAC dramatically improves precision-recall of fitness genes across the genome, allowing us to generate the most comprehensive list of cell fitness genes yet assembled for *Drosophila*. *Drosophila* fitness genes show strong correlation with human fitness genes and underscore the effects of paralogs on gene essentiality. We also perform a resistance screen to proaerolysin, a glycosyl-phosphatidylinositol-(GPI)-binding pore-forming toxin, retrieving 18/23 expected and one previously uncharacterized GPI synthesis gene. We also demonstrate that an IntAC sublibrary enables precise positive selection of a transporter under solute overload. IntAC represents a straightforward enhancement to existing *Drosophila* CRISPR screening methods, dramatically increasing accuracy, and might also be broadly applicable to virus-free CRISPR screens in other cell and species types.

CRISPR-Cas9 screens have become a critical tool for systematically mapping genotype-to-phenotype relationships[1]. In mammalian systems, these screens have been used to uncover genes essential for cell viability, immune modulation, and responses to therapeutic agents, leading to key insights into cancer biology and genetic diseases.

However, adapting these powerful screens to insect models like *Drosophila melanogaster* and *Anopheles gambiae* has presented significant challenges, particularly due to the lack of an efficient retroviral induction system in these organisms. We previously showed that site-specific recombination is an efficient alternative to retroviral vector

[1]Department of Genetics, Harvard Medical School, Boston, MA 02115, USA. [2]Howard Hughes Medical Institute, Boston, MA 02115, USA.
✉e-mail: ram@genetics.med.harvard.edu; perrimon@genetics.med.harvard.edu

delivery for CRISPR screens and performed the first pooled CRISPR screens in insect cells[2,3]. The approach has provided insights into essential genes and signaling pathways, hormone transport, and toxin tropism and trafficking, and is compatible with alternative CRISPR types such as CRISPR activation[2,4–6]. Nevertheless, quantification of the precision-recall in our screening platform showed that it under-performed as compared with optimized human cell line screens, such that critical genes regulating these functions could be obscured by error or missed altogether[2].

One of the primary drawbacks in our transfection-based CRISPR screen platforms stems from the timing and control of Cas9 activity. In our previous screens, cells stably expressing Cas9 were transfected with plasmids expressing sgRNAs[2]. This poses a problem because multiple sgRNAs can be expressed from free plasmids in a single cell prior to integration into the genome, leading to genome editing, but non-integrated sgRNA plasmids decay through cell divisions, breaking the relationship between the sgRNA causing an observed phenotype and the integrated sgRNA that is detected in the next-generation sequencing (NGS) step. Therefore, we reasoned that precise temporal control of Cas9 activity, i.e., silencing activity prior to sgRNA integration, would improve screen resolution.

Strategies used to achieve temporal Cas9 control face the limitations of leakiness or low reactivation efficiency. In *Drosophila*, we previously tried treatment-inducible expression of Cas9 using a metal-inducible metallothionine promoter; however, expression of the promoter under normal media conditions was too great to suppress Cas9 editing in the uninduced state[2]. Tighter strategies for drug-controlled expression are lacking in *Drosophila*. Conversely, intein-based drug-inducible Cas9, while fully repressed in the non-induction condition, shows low reactivation efficiency[2,7]. Additional drug induction systems for Cas9 using TMP or rapamycin[8–10] might work in *Drosophila* but have not yet been reported. Optogenetic systems that allow for light-controlled Cas9 may offer precise control but are difficult to implement in large-scale cell culture screens[11,12]. Introducing Cas9 in a second transfection step after sgRNA integration, the favored method in virus-free screens to date[13–15], would improve timing but introduces new challenges, such as the expense, potential for genetic drift incurred by a second large-scale transfection, and the difficulty in selecting Cas9-transfected cells for sequencing and subsequent challenge treatments while avoiding cells that received sgRNAs but not Cas9 (which would elevate noise in the NGS step). Finally, 'Guide-swap', where Cas9 is electroporated into cell pools expressing sgRNAs with a dummy guide that is then 'swapped' for the cell-encoded guide, allows precise timing control but is expensive and challenging to scale for large screens[16].

To address these issues, we develop IntAC (integrase with anti-CRISPR), a system that integrates sgRNA libraries with built-in temporal control over Cas9 activity in a single transfection step. By expressing the anti-CRISPR protein AcrIIA4[17–19] via a plasmid at the time of sgRNA transfection, we delay Cas9-mediated genome editing until after stable sgRNA integration has occurred. This method circumvents the need for sub-optimal induction systems or additional transfection steps, providing a straightforward and scalable solution. Another important improvement is the use of the strong *dU6:3* promoter[20]. Previous *Drosophila* CRISPR screens used the weaker *dU6:2* promoter since lowering sgRNA levels helped maintain the phenotype-sgRNA linkage[2]. The use of anti-CRISPR allowed us to use the strong *dU6:3* promoter to drive sgRNA expression in the pooled library cells, leading to higher screen resolution and improved detection of high-confidence fitness genes. Fitness genes are defined as those genes needed for optimal cell proliferation[21]. We additionally test the method in positive selection CRISPR screens. In a genome-wide screen for proaerolysin (PA) resistance, we identify 18/23 predicted gene orthologs underlying PA sensitivity by enabling the synthesis of GPI anchors, the major receptors of aerolysins. In *Drosophila* we additionally

identify a reliance on complex N-glycans and the importance of a small open reading frame (ORF) gene, which we show is a component of the *Drosophila* GPI anchor synthesis pathway. Finally, we show that the IntAC method retains precision when used with a small library of solute transporter gene-targeting sgRNAs when performing screens for solute overload.

## Results

### Transient inactivation of Cas9 by anti-CRISPR

In our efforts to enhance screen resolution, we began by addressing the possibility that multiple sgRNAs are expressed in the same cell. In our previous approach, hereafter referred to as "version 1" (v.1), we transfected Cas9-expressing cells with a plasmid library of attB-flanked sgRNAs driven by the weak *dU6:2* promoter into cells that have an attP integration site (Fig. 1A). The success of our previous CRISPR screens using this approach suggests that the sgRNA that integrated into the attP site predominantly led to the edit recorded at the end of the assay[2–4,6], but it is likely that additional edits were made during the initial period following transfection. Consistent with this, pilot studies in which we increased the sgRNA levels in a small library using the stronger *dU6:3* promoter[20] led to instability of the library and death of a large fraction of transfected cells, possibly due to many cells acquiring early edits in essential genes and/or hyperactivation of the DNA damage response due to individual cells receiving too many double-strand breaks. To limit Cas9 activity until sgRNA cassette integration and loss of non-integrated sgRNA plasmids, we devised the integrase with anti-CRISPR (IntAC) approach. During library transfection, we included a plasmid encoding phage *φC31* integrase linked to the anti-CRISPR *AcrIIA4* by a T2A self-cleaving peptide. AcrIIA4 is the most potent known protein inhibitor of Cas9 activity[17–19]. This protein binds to the Cas9–sgRNA complex mimicking DNA, obstructing the PAM recognition site, and impeding the necessary conformational changes. Cas9 activity is thereby suppressed during the initial transfection period and then automatically restored when plasmids decay through a natural process of dilution during cell divisions. By this point, one sgRNA at random would have integrated into each attP site, and thus, genomic edits made after anti-CRISPR activity is lost should be restricted to integrated sgRNAs (Fig. 1B). Furthermore, because the IntAC approach requires only one transient transfection step and uses the highly active constitutive *Streptococcus pyogenes* Cas9, the approach improves the accuracy of genotype–phenotype mapping.

To test whether the IntAC approach suppresses cutting by Cas9, we drove a highly efficient sgRNA targeting the *Rho1* gene with the *dU6:3* promoter and monitored editing over time using the T7 endo-nuclease I-sensitivity assay. Rho1 knockout cells initially become large[22] due to a failure in cytokinesis and die by an unknown process after several weeks. Cells transfected with *pLib6.6/Rho1* alone or alongside a second non-integrating vector expressing *φC31* integrase ('Int') or *φC31-integrase-T2A-AcrIIA4* ('IntAC') were sampled at 10, 14, or 18 days after transfection. Editing reached a maximum (of ~75%) 10–14 days after transfection when the sgRNA vector was transfected along with Int. However, the presence of the IntAC plasmid imparted a clear delay; for the IntAC population, editing efficiency was initially low on days 10 and 14 but then reached nearly the level of the Int population (~65%) on day 18 (Fig. 1C). As expected, the visible phenotype of *Rho1* knockout, i.e., large cells[2,22], also exhibited a delay (Fig. 1D). This indicates that anti-CRISPR co-transfection imparts a delay to editing that is reversible in most cells when the plasmid encoding anti-CRISPR decays away.

### Version 2 (v.2) CRISPR screening library: optimization of sgRNA library using machine-learning and use of the strong *dU6:3* promoter

In addition to the level of editing reagents delivered to cells, previous studies have shown that optimizing sgRNA parameters through machine-learning analysis of a previous screen can also improve

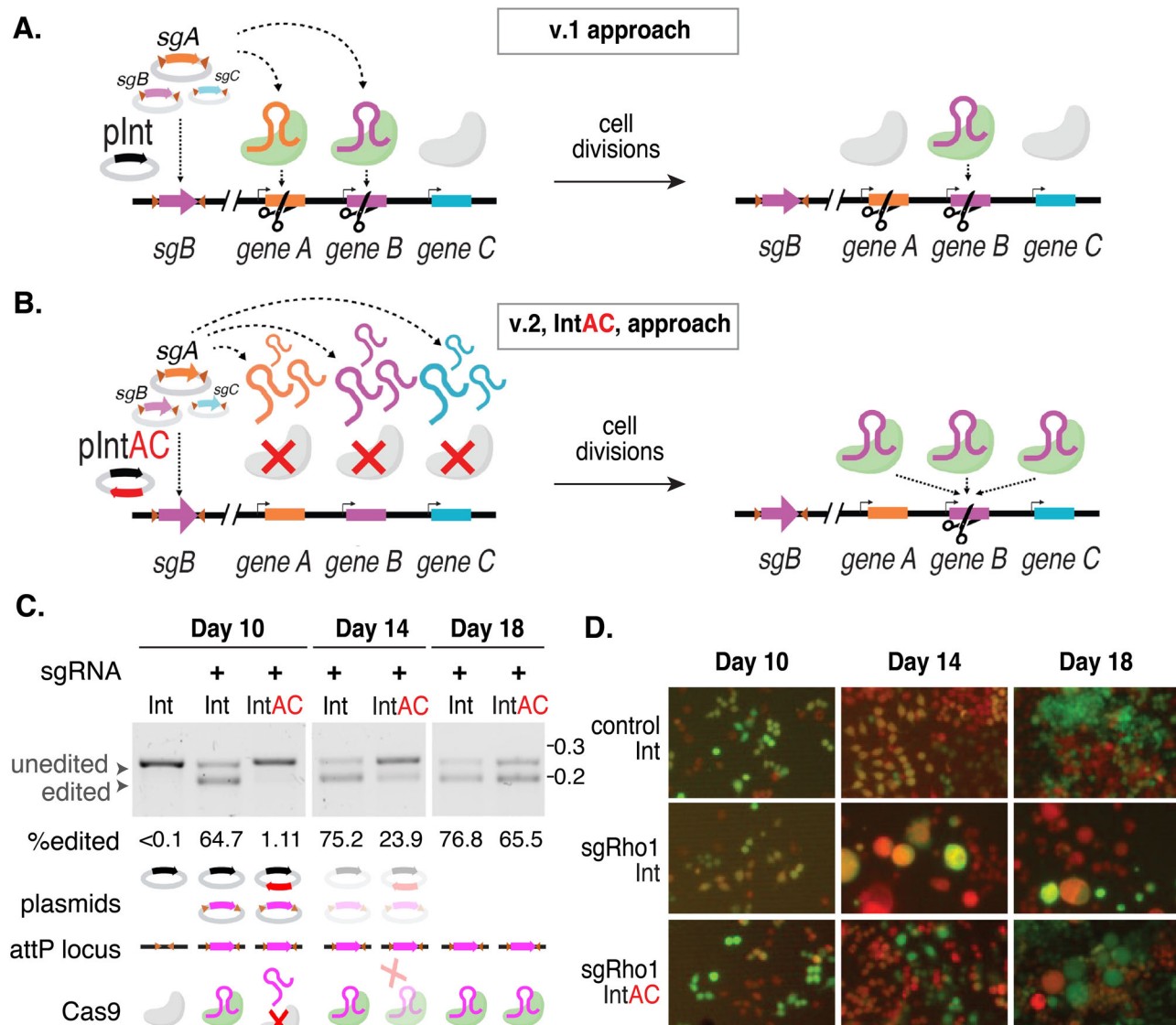

**Fig. 1 | Schematic of the IntAC system and validation of transient inhibition by anti-CRISPR in *Drosophila* cells.** Schematic representation of the current CRISPR screening system compared with the IntAC system. **A** In the v.1 approach, sgRNAs are active upon pooled transfection and one of these at random integrates into the cell's genome via φC31 integration. Following cell divisions and the loss of transiently transfected plasmids, one sgRNA, illustrated by *sgA* (purple), is found in the cell's genome, but undesired cutting by other sgRNAs may have occurred, illustrated by *sgB* (orange). This is tolerable due to the weaker promoter for sgRNAs (*U6:2*). Elements created in BioRender. Viswanatha, R. (2025) https://BioRender.com/0w8vn0z. **B** In the v.2 approach, anti-CRISPR (red) is expressed, initially inactivating Cas9 and preventing CRISPR, while one sgRNA is still integrated into the cell's genome using φC31 integration. Later, following cell divisions, transiently transfected sgRNAs are lost, and the integrated sgRNA alone is active in the cell. We also used the stronger U6:3 promoter in v.2 to express more sgRNA. pInt = φC31 Integrase expression plasmid. pIntAC = φC31-T2A-AcrIIa4 expression plasmid. **C** T7 endonuclease I-sensitivity assay, showing the edited versus unedited allele targeted by a CRISPR sgRNA measured over time in cells transfected as indicated. Cell populations transfected with IntAC exhibited a clear early delay in editing, with editing efficiency returning to nearly that of controls by day 18. Quantification of editing (edited band intensity divided by the sum of the intensities of the edited and unedited bands). Representative of three experimental replicates. Molecular weight markers are given in kilobases. Created in BioRender. Viswanatha, R. (2025) https://BioRender.com/zmb3fwf. **D** Phenotypic validation of IntAC-delayed editing. Cells with *Rho1* suppression, characterized by large cell morphology, were observed later in the IntAC population compared to controls, confirming the temporal delay in Cas9 activity. Red = mCherry in *Drosophila* S2R+ derivative PT5 (NPT005; DGRC #229) cells; Green = free GFP expression from sgRNA plasmid (*pLib6.6/sgRho1*) which additionally encodes *Actin* promoter-driven GFP.

resolution[23]. We applied this approach, using a machine learning analysis of our previous (v.1) library screen data to develop a new ruleset for sgRNA design based on an analysis of dinucleotide position and other physical parameters, as previously described and implemented in our FlyRNAi suite of online tools[3,24] (Fig. 2A). We then constructed a new (v.2) library of 92,795 sgRNAs targeting all *Drosophila melanogaster* genes as annotated by FlyBase version 6, with a modal number of 6 sgRNAs per gene, and expressed the sgRNAs under the control of the strong *dU6:3* promoter. The library was transfected along with IntAC in two biological replicates, passaged continuously for approximately

two months, and then sequenced (Fig. 2B). To evaluate the library and IntAC approach, we first compared the fate of 17,948 sgRNA sequences present in both the v.1 and v.2 libraries (Fig. 2C, Supplementary Data 1). In an analysis of this sgRNA set at the sgRNA level, we found that significantly more sgRNAs consistently dropped out of the pool in both replicates of v.2 as compared with v.1, suggesting that higher sgRNA levels alone led to more mutations in the v.2 cell pool as compared with v.1 (Fig. 2D). To determine whether this higher level of sgRNA 'dropout' at the sgRNA level is reflected in genes detected as essential for cell fitness, the sgRNA log2 fold-changes were processed

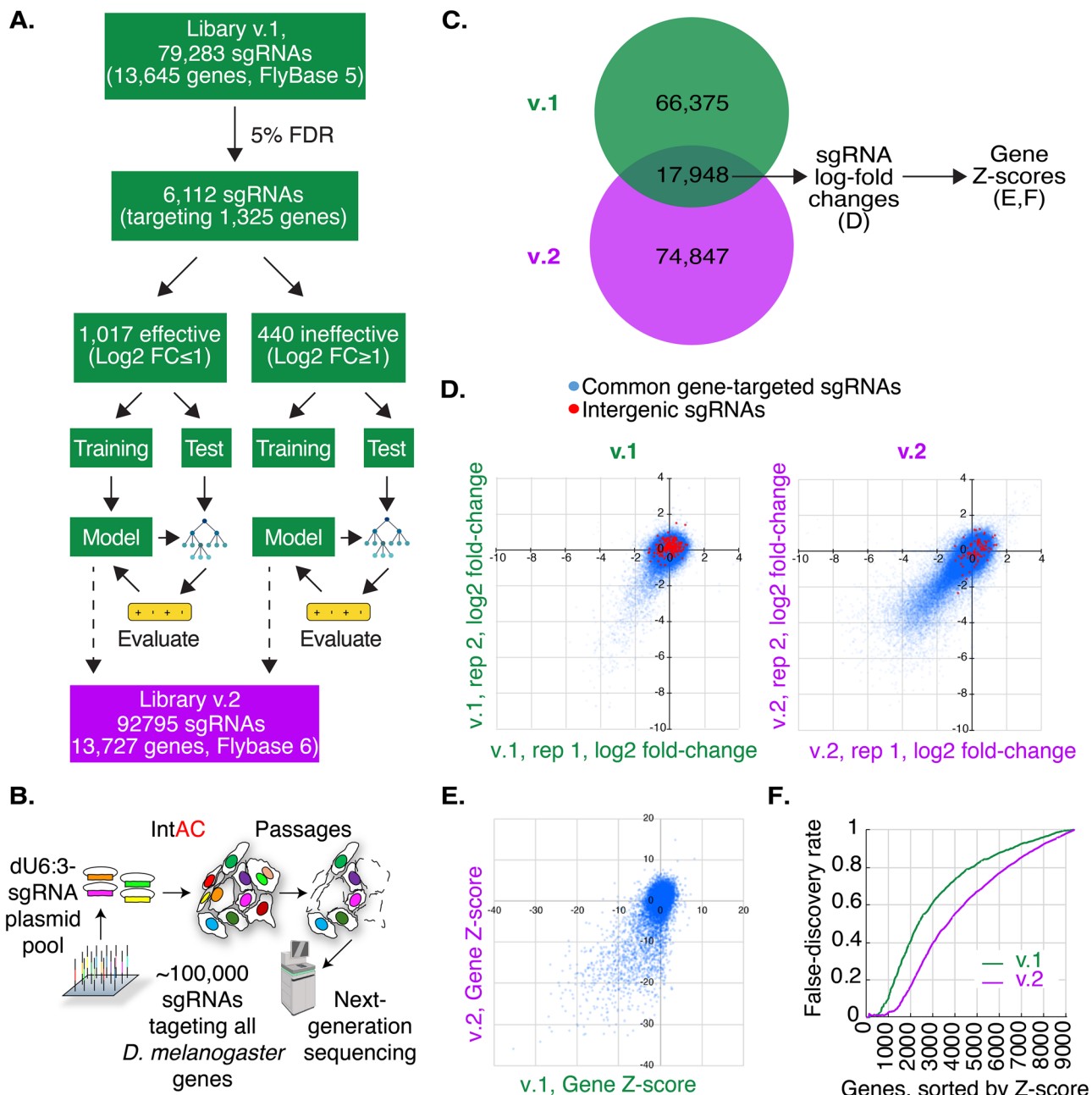

**Fig. 2 | IntAC and *dU6:3* drive much of the performance increase in an updated *Drosophila* CRISPR screening library. A** Overview of machine-learning optimization of a new ('v.2') genome-wide CRISPR screening library for Drosophila. Elements created in BioRender. Viswanatha, R. (2025) https://BioRender.com/y0pxkoa.
**B** Schematic representation of transfection of the v.2 library. 92,795 attB-flanked sgRNAs targeting *Drosophila melanogaster* genes were transfected together. sgRNA expression was driven by the strong *dU6:3* promoter, and the library was co-transfected with the IntAC system (a plasmid expressing φC31 integrase and AcrIIa4) into cells. Cells were passaged for approximately two months, and then CRISPR sgRNA distributions were assessed by next-generation sequencing. Elements created in BioRender. Viswanatha, R. (2025) https://BioRender.com/2v5d13e.

**C** sgRNAs common to v.1 and v.2 libraries were analyzed to isolate the effect of IntAC and *dU6:3*. **D** Comparison of 17,948 sgRNAs shows dramatically increased, consistent dropout of sgRNAs in v.2 relative to v.1. Plot shows normalized log2 fold-change of gene-targeting (blue) or intergenic (red) sgRNAs. **E** Gene-level analysis of the sgRNA log2-fold changes in **D** shows that the increased sgRNA dropouts translate to increased fitness gene calls (genes with negative *Z*-scores). **F** Cumulative distribution of false-discovery of fitness genes (genes with low expression, FPMK < 1) from v.1 (green) or v.2 (purple) gene *Z*-scores in **E** shows that IntAC and *dU6:3* increase the detection of fitness genes. This figure was created in part with Biorender.

using maximum likelihood estimation (MLE in MAGeCK[25],) to obtain gene-level *Z*-scores. Negative *Z*-scores indicate a stronger likelihood of essentiality for fitness. Using the rate of discovery of non-expressed genes (i.e., genes with an FPKM < 1[26]), which cannot have a role in cell fitness, to define a false discovery rate[2], we found that use of anti-CRISPR and the *dU63* promoter more than doubled the number of bona fide fitness genes that could be assigned with an

error rate of 5% (Fig. 2E, F, Supplementary Data 1). These results suggest that much of the improvement between v.1 and v.2 is driven by higher sgRNA expression levels via the *dU6:3* promoter and the improved 1-to-1 relationship between detected sgRNA and edited genes enabled by the use of anti-CRISPR. We hypothesize that sgRNA design optimization using machine learning had a relatively minor impact.

## IntAC screens exhibit dramatically improved fitness gene assignment

We conducted genome-wide fitness screens using the v.2 screening library with or without IntAC, and compared the results to v.1 screen data[2] and genome-wide arrayed RNAi screens[27]. For each method, we binned genes based on their predicted likelihood of being fitness genes (from low to high MLE $Z$-score). Because genes required for cell fitness must also be expressed as RNA, we looked at the RNA expression of genes in each bin. As expected, genes in lower bins were highly enriched for expressed genes and had very few non-expressed genes, unlike intermediate or high bins. The strength of this enrichment indicates how many fitness genes were uncovered by each method and thus serves as an indicator of screen quality. Compared to v.1, the v.2 screen with IntAC had more bins enriched in expressed genes, indicating a higher number of fitness genes detected and improved screen quality (Fig. 3A). Furthermore, running the v.2 screen without anti-CRISPR led to a marked decrease in fitness genes detected and therefore lower screen quality (Fig. 3A). This finding illustrates why anti-CRISPR inhibition is crucial when using the v.2 library and strongly supports our hypothesis (and pilot data) suggesting that early, unregulated Cas9 activity in the absence of anti-CRISPR diminishes screen precision, especially when paired with elevated sgRNA levels driven by the stronger *dU6:3* promoter.

To determine whether known essential genes were detected as fitness genes, we asked whether core components of the spliceosome, ribosome, or proteasome (as assigned by KEGG) were detected in gene-level analysis of v.1 or v.2, IntAC, screens (Supplementary Data 1). While these gene sets were statistically enriched in v.1 screens, the majority of their component genes were missed. Conversely, in v.2 IntAC screens, most genes in each gene set were detected (Fig. 3B). Moreover, as expected, comparison of the two gene sets shows that the v.1 fitness gene set is a subset of the v.2 fitness gene set (Supplementary Fig. 1). To quantify the rate of recall assuming that the KEGG assignment is correct, we assigned the genes in each KEGG gene set as the set of true positives. Using non-expressed genes (FPKM < 1) as the set of false positives, we estimated a precision-recall rate. The results show that the recall at 5% is 35–60% for v.1 and 90–95% with v.2 (Fig. 3C). Thus, using the improved v.2 IntAC CRISPR screen approach, we identified the most complete and accurate set of cell fitness genes yet assembled for *Drosophila* or any invertebrate, likely capturing 90–95% of all fitness genes with 5% error. Importantly, we note that this approach and the library have now been applied independently by multiple members of our group and resulted in comparable precision-recall rates (Supplementary Fig. 2A,B).

## An updated comparison of *Drosophila* and human cell fitness genes

We next compared the high-resolution *Drosophila* cell fitness screen data to data from human cell line CRISPR screens. First, we determined the *Drosophila* orthologs of human 'core essential genes' (CEG2) assembled from human CRISPR and RNAi screens[28]. The majority of these genes are present in our *Drosophila* v.2 CRISPR screen (~90% overlap, allowing 5% false-discovery in the *Drosophila* v.2 CRISPR screen data, Supplementary Fig. 3), but CEG2 orthologs only account for ~17% of the *Drosophila* v.2 gen-eset. We reasoned that some of the remaining fitness genes in *Drosophila* cells might reflect genes that were missed due to masking of essential gene functions in human CRISPR screens. Whole-genome duplication (WGD) events create paralogs with the potential for redundant functions; invertebrates underwent fewer WGD events than humans[29]. To address this, we asked whether our CRISPR screen data correlates with human cell line CRISPR screens for *Drosophila* genes with a 1-to-1 relationship with human genes, and whether our findings for "1-to-many" genes

lend further support to the idea that increased genetic redundancy in human cells masks detection of essential gene functions. To do this, we first retrieved all human gene orthologs for all *Drosophila* genes using the DIOPT approach[30] and then computed the first quartile of all Cas9-mediated Essentiality Ranking Evaluator of Specificity (CERES) scores ('Q1 CERES') to represent a consensus fitness score across all human cell lines for every gene tested in the human Dependency Map (DepMap)[31]. To map orthologs, we used a DIOPT score of 6 or greater, resulting in 4380 1-to-1 orthologs and 2206 1-to-many orthologs. We observed a strong correlation for 1-to-1 orthologs in both species ($R^2 = 0.41$, Fig. 4A). However, consistent with expectation, this correlation fell significantly for genes that have two or more orthologs in humans ($R^2 = 0.18$), even when we intentionally counter-biased the analysis by choosing the most essential human ortholog to represent the human gene in the pair (Fig. 4B). This finding supports the idea that redundancy due to paralogs is an important factor in CRISPR screen data interpretation, and highlights paralog groups in human cells that do not score positive in screens but collectively represent an essential function, as their unique *Drosophila* ortholog is required for optimal fitness. A list of 123 gene groups matching these criteria is provided in Supplementary Data 2.

## Positive selection screening using a genome-wide IntAC screen

To further evaluate the utility of the IntAC system in positive selection screens, we conducted a genome-wide screen for resistance to proaerolysin (PA), a beta-sheet pore-forming toxin (β-PFT) that utilizes glycosylphosphatidylinositol (GPI)-anchored proteins on the cell surface to homoheptamerize, forming a pore through which anions flow out of cells, ultimately leading to cell death[32]. PA is produced in an inactivated form by *Aeromonas hydrophila*, requiring cleavage by a secreted protease in order to become active[33]. Previous studies using mammalian cells have found that mutating the GPI anchor synthesis pathway or the complex N-glycosylation pathway results in reduced or eliminated binding and killing by PA[34–36]. Resistance factors have been functionally mapped in a genome-scale screen[34] and a focused library of mutants in the GPI anchor synthesis pathway[35] (Fig. 5A).

To conduct a PA resistance screen in *Drosophila* cells, we first determined that *Drosophila* S2R+ cells are sensitive to PA in the low nanomolar range. We transfected the v.2 genome-wide sgRNA library using the IntAC approach and treated cells with a lethal dose of PA (2 nM). Following 3 weeks of selection, populations developed 10–100-fold resistance (Supplementary Fig. 4A), and we sequenced their sgRNAs. Analysis of the enriched sgRNAs revealed genes involved in the GPI anchor synthesis and complex N-glycosylation pathways, reproducing previous findings in mammalian cells (Fig. 5B; Supplementary Data 1). We next superimposed our gene hits upon the established linear GPI anchor synthesis pathway (Fig. 5C). A comparison with two previous human cell studies[34,35] showed that 18/23 expected orthologs were significantly enriched (top 43) in the IntAC screen (Fig. 5C). Among the five missed genes, one of the two orthologs of *PIGW*, *PIG-Wa*, was nonetheless partially enriched (gene # 177 out of the 13,727 genes targeted in the library). Four expected orthologs (*PIG-F*, *-N*, *-Q*, and *PGAP5*) were not enriched at all, without a single highly enriched sgRNA among the six sgRNA targeting each gene. Among these, only *PGAP5* has been previously shown to regulate GPI anchor synthesis in *Drosophila*[37], whereas the others have not been experimentally tested. These results strongly indicate that the IntAC resistance screen shows an alignment between the genetic requirements for PA killing between mammalian and *Drosophila* cells. Moreover, the majority (>75%) of expected components were detected. The importance of the missing components for PA killing in *Drosophila* remains unclear.

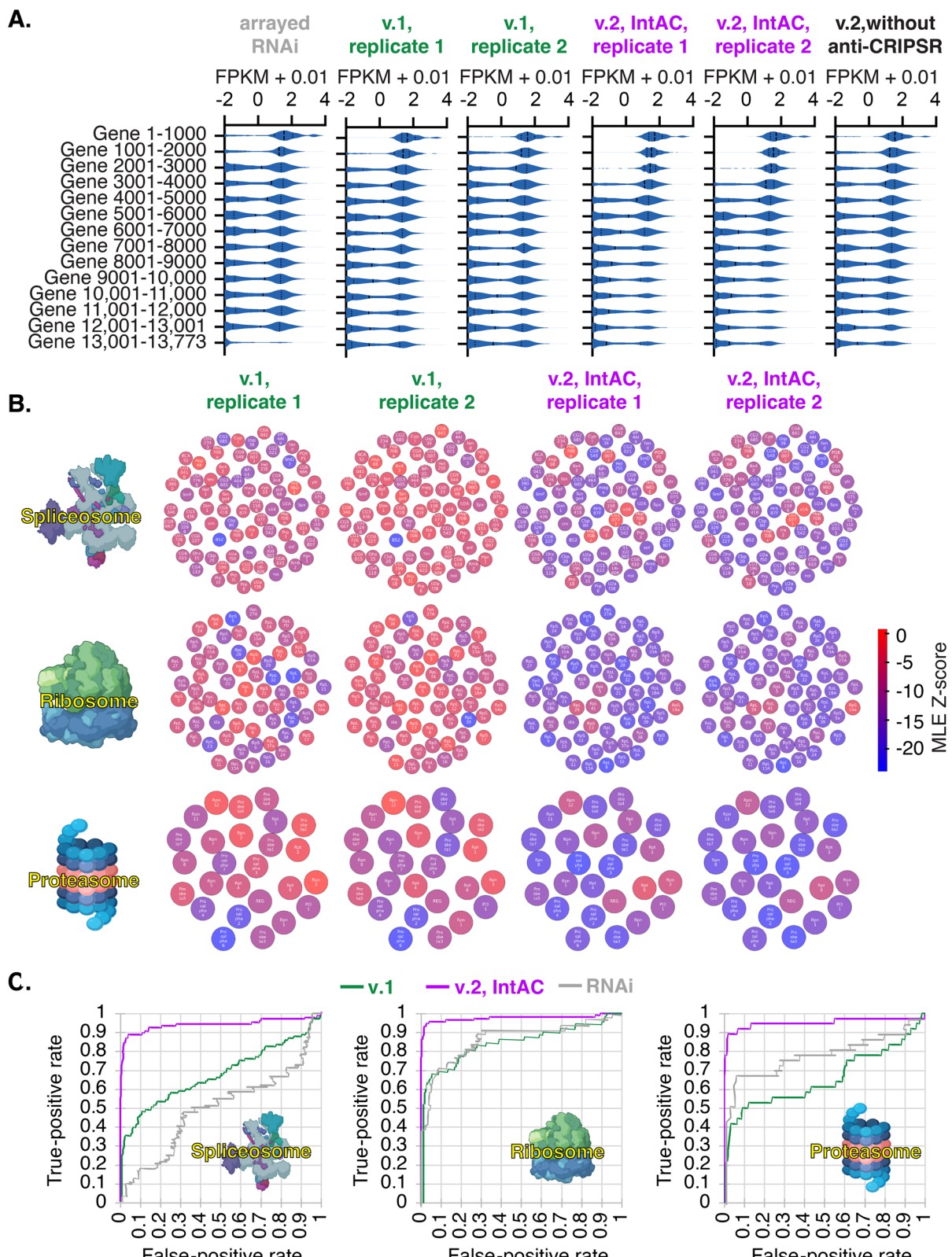

Among our enriched genes, we were surprised to identify 4/4 sgRNAs targeting an uncharacterized small open reading frame (ORF), *CG46311* (Fig. 5C). Its coding region is found inside a larger gene, *Tpst*, which has a role in membrane trafficking and therefore could plausibly be responsible for PA resistance. However, 6/6 sgRNAs directly targeting the coding sequence of *Tpst* did not show significant enrichment (Fig. 6A). Moreover, cells expressing sgRNAs against CG46311 were

partially resistant to PA, as demonstrated by a Cell Titer Glo viability assay (Fig. 6B). To determine whether CG46311 acts in the GPI anchor synthesis pathway, we transfected cells with HsCD58[GPI], a canonical tracer of the pathway[38]. Similar to PIG-A knockout cells, CG6311 knockout cells displayed internal but not membrane-targeted GFP-HsCD58[GPI] (Supplementary Fig. 4B). Moreover, GFP-HsCD58[GPI] was detectable on the extracellular surface of fixed but non-permeabilized CG46311 KO

**Fig. 3 | Nearly complete genome-wide cell fitness gene assignment by IntAC. A** Expression level of fitness genes (FPKM + 0.01) assigned by genome-wide arrayed RNAi screens[27], v.1 CRISPR screens, v.2 IntAC CRISPR screens, or v.2 CRISPR screens without anti-CRISPR. Genes arranged from low to high maximum likelihood estimation $Z$-scores and binned in groups of 1000 genes. Since only expressed genes can be fitness genes, v.2 IntAC screens lead to superior fitness gene assignment, and leaving out anti-CRISPR severely diminishes fitness gene identification. **B** Detection of essential genes across components of essential cell structures—spliceosome, ribosome, and proteasome—in v.1 and v.2 screens. While these genes were enriched in v.1 screens, a substantial portion was missed. In contrast, the majority of the core components were detected in the v.2 IntAC screens, illustrating the improved

sensitivity of the IntAC system for detecting essential cellular processes. The color of each component reflects its gene $Z$-score. **C** Precision-recall curves for fitness gene assignment in v.1 screens (green), v.2 screens (purple), or genome-wide arrayed RNAi screens (gray) using components in **B** as true-positives and non-expressed genes as false-positives. v.2 IntAC screens exhibited a recall rate of 90–95% for essential genes, compared to 35–60% in v.1. This marked improvement in recall demonstrates that the IntAC system yields more comprehensive and accurate identification of fitness genes across the genome. Genome-wide RNAi screen data is from ref. 27. Elements created in BioRender. Viswanatha, R. (2025) https://BioRender.com/orvd36w.

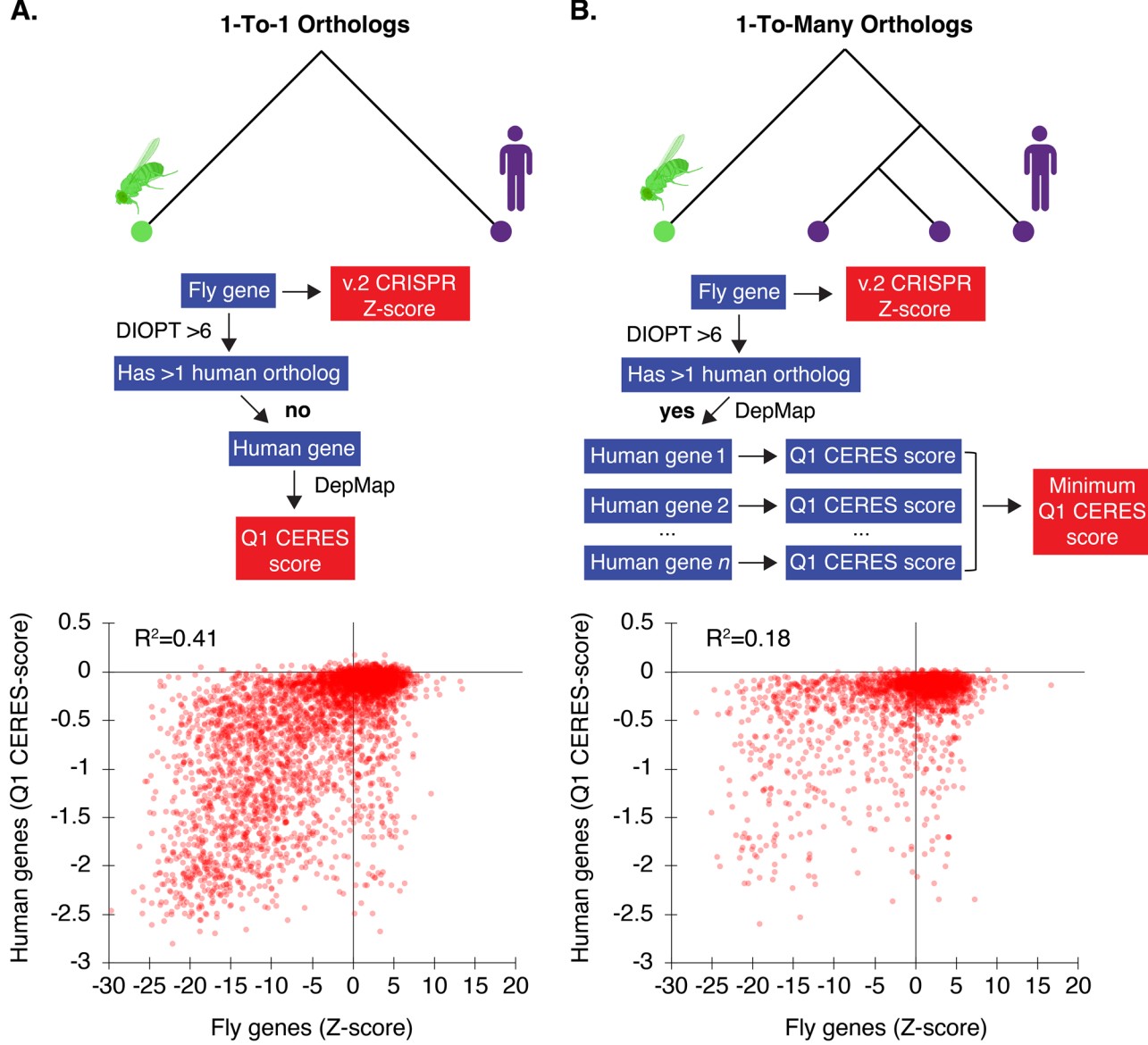

**Fig. 4 | Correlation between *Drosophila* fitness genes and human orthologs in IntAC screens. A** For genes with a 1-to-1 ortholog relationship between *Drosophila* and humans, a strong correlation ($R^2 = 0.41$) was observed, suggesting high conservation of fitness genes across species. Schematic of processing steps comparing *Drosophila* orthologs to their predicted human orthologs (assessed by a DIOPT score > 6, using the DRSC Integrative Ortholog Prediction Tool[30], which integrates multiple methods of assessing gene orthology from gene sequence). Genes with a 1-to-1 relationship were retained for analysis, and the first-quartile (Q1) of their Cas9-mediated essentiality ranking evaluator of specificity (CERES) scores was plotted

across all DepMap data (23Q4)[31]. **B** Any gene found to have more than 1 ortholog at a DIOPT score of >6 was considered to have a 1-to-many relationship (*Drosophila* genes with multiple human orthologs). In these cases, the correlation with fitness scores dropped significantly ($R^2 = 0.18$). Schematic showing processing steps. When multiple human orthologs had a DIOPT score > 6, the ortholog with the lowest CERES score was chosen in order to counter-bias the data against selecting an inactive paralog. This result suggests that paralog redundancy in humans can mask the essentiality of certain genes. Elements created in BioRender. Viswanatha, R. (2025) https://BioRender.com/lmkhxtm.

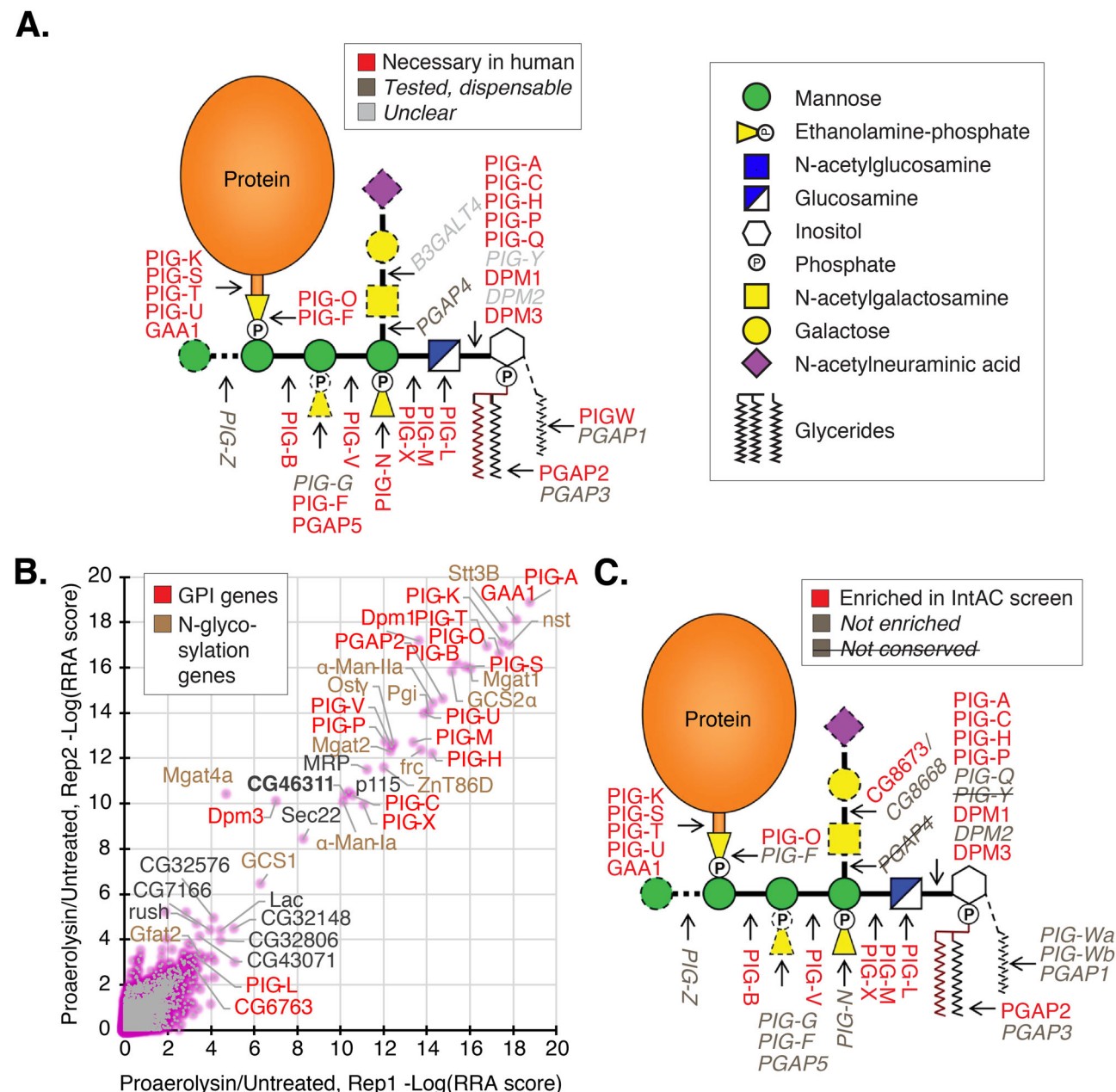

**Fig. 5 | Genome-wide positive selection of GPI-anchor-synthesis- and N-glycosylation-defective cells in a proaerolysin (PA)-resistance screen.**
**A** Schematic of the eukaryotic GPI anchor synthesis pathway highlighting genes shown to be necessary (red) or dispensable/unclear (dark gray/light gray) for human cell sensitivity to aerolysin[34,35]. **B** A genome-wide IntAC PA-resistance screen identifies genes in the GPI anchor synthesis (red) and N-glycosylation (brown) pathways, confirming the results of past studies in mammalian cells[34–36]. Plotted values are -Log of MAGeCK[25] robust rank aggregation (RRA) scores from two

independent replicates of PA treatment compared to sgRNA distribution in a common untreated cell population. **C** The orthologs of the genes identified in the PA resistance screen from **B** are placed within the GPI anchor synthesis pathway, highlighting genes that are enriched in the *Drosophila* IntAC screen (red) versus not enriched or not annotated as conserved (gray or gray strikethrough). Figure adapted from Liu et al. A knockout cell library of GPI biosynthetic genes for functional studies of GPI-anchored proteins, 2021 Communications Biology[35] (CC-BY-4.0). http://creativecommons.org/licenses/by/4.0/).

cells, and this was rescuable by re-expression of *CG46311* cDNA (Fig. 6C). The rescue of GFP-HsCD58$^{GPI}$ trafficking by overexpression of CG46311 cDNA further shows *CG46311* but not *Tpst* underlies the GPI anchor defect. The strong dependence on CG46311 for GFP-GPI localization led us to ask whether it is a component of the conserved Glycosylphosphatidylinositol-N-acetylglucosaminyltransferase (GnT) complex, which catalyzes the first step in GPI anchor synthesis in the endoplasmic reticulum. The mammalian GnT complex has been described to contain seven proteins: PIG-A, PIG-C, PIG-P, PIG-H, PIG-Q, DPM-2, and PIG-Y[39]. Homologs of 6/7 genes are well represented in

*Drosophila*, the exception being *PIG-Y*. While *CG46311* and *PIG-Y* show low sequence homology (27.5% protein identity; Supplementary Fig. 4C), both are largely alpha-helical in predicted secondary structure. Remarkably, Alphafold-Multimer modeling predicts a strong interaction between CG46311 and multiple proteins within the *Drosophila* GnT complex. Scoring metrics (predicted aligned error, PAE, as well as interface predicted Template Modeling, ipTM) suggest that CG46311 engages in a high confidence interaction with PIG-A (0.816 ipTM), a moderate confidence interaction with PIG-P (0.728 ipTM), and low confidence interactions with PIG-C, PIG-Q, and Dpm2, mirroring the

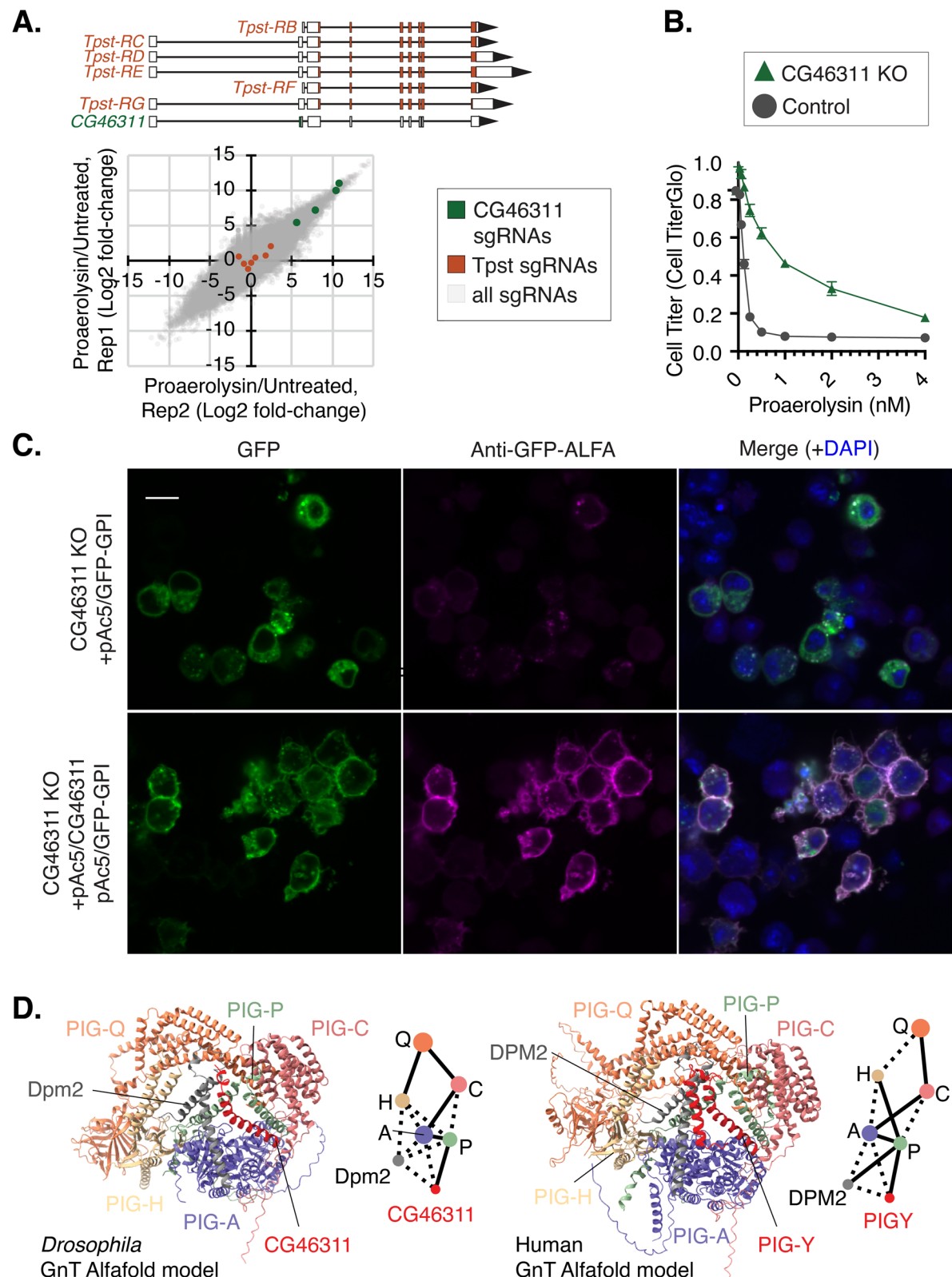

predicted interaction strengths for PIG-Y within the human GnT complex (Fig. 6D; Supplementary Fig. 4D). Moreover, CG46311 is predicted to bind within the *Drosophila* GnT complex in a position similar to that of human PIG-Y within the human GnT complex (Fig. 6D). Therefore, we conclude that CG46311 is a component of the GPI anchor synthesis pathway in *Drosophila* and likely represents the "missing" PIG-Y ortholog. These findings demonstrate that the IntAC system can be effectively

utilized in positive selection screens to identify genes involved in toxin susceptibility.

## Positive selection of transporters mediating solute overload using IntAC screens

Next, we asked how the IntAC method performs in small, focused sublibraries of sgRNAs. Matching solutes to their transporters is important

**Fig. 6 | A small open reading frame (smORF) gene is identified as a novel GPI anchor synthesis component in *Drosophila*. A** sgRNAs targeting a small ORF (present in the v.2 library but not v.1), CG46311 (green), expressed within the *Tpst* locus, provides resistance to PA; sgRNAs targeting *Tpst* coding regions (brown) are not significantly enriched. All other sgRNAs in the screen are indicated in gray. **B** Cell Titer Glo viability assay with varying dosage of PA demonstrating that CG46311 knockout (KO) cells (green) display partial resistance to PA compared with control cells (gray). Error bars are the standard deviation from three replicates. **C** Detection of surface-exposed GFP-tagged GPI reporter (GFP-HsCD58^GPI) in CG46311 KO cells fixed but not permeabilized, and rescue by co-transfection of CG46311 cDNA. All transfected cDNAs are driven by the strong *Drosophila ActinSC* promoter (pAc). Scale bar = 20 μm. Representative of three experimental replicates. **D** Alphafold-Multimer model (Alphafold3) of *Drosophila* or Human GnT complex, highlighting the 7 polypeptide chains in each model. Interface view (ChimeraX) showing the similarity in overall orientation of the complexes. Solid lines represent large solvent accessible interfaces, whereas dashed lines represent smaller solvent accessible interfaces. Line lengths are arbitrary.

to understand how cells manage the intake, export, and compartmentalization of solutes and how this process goes awry in metabolic disorders[40]. In *Drosophila*, approximately one-third of predicted transporters are annotated as "CGs" (computed genes) without documented phenotypes[30]. To provide greater insight into these genes, we set out to target them with higher resolution than the genome-wide library, i.e., using more sgRNAs per gene. We prepared a new sub-library containing 10 sgRNAs per gene for 389 solute carrier genes, selecting only those expressed in *Drosophila* S2R+ cells. We then transfected cells using the IntAC approach (Fig. 7A). In a pilot panel of solutes, we discovered that the nucleoside cytidine was extremely toxic to cells at a concentration of 250 μM. We then performed two biological replicates of a screen for resistance to cytidine overload. The genes *Ent1* and *Ent2* are thought to mediate nucleoside uptake in *Drosophila*[41,42], but only *Ent2* is expressed in S2R+ cells[43]. Following selection, *Ent2* sgRNAs predominated reads, and non-*Ent2* sgRNAs (likely sgRNAs that were co-transfected in the same cells and became enriched along with Ent2 sgRNAs, i.e., 'passengers') emerged only after the seventh *Ent2* sgRNA in replicate 1 or the fifth *Ent2* sgRNA in replicate 2 in the ranked dataset (Fig. 7B, C; Supplementary Data 1). The precision of this selection again suggests that IntAC results in cells edited primarily by integrated sgRNAs, even when, in this case, we combine it not only with the use of a higher strength promoter *dU6:3*, but also use of a positive selection assay and a smaller sgRNA library. Thus, the IntAC approach is extensible to positive selection screens and focused sub-libraries.

## Discussion

Pooled CRISPR screening is the state-of-the-art genetic screening approach in *Drosophila* cells[2]. *Drosophila* CRISPR screens provide significantly improved detection of fitness genes compared to arrayed RNAi screens[2], so revisiting previous RNAi screens using CRISPR screening could yield additional insights. The method has already provided insights into essential genes and signaling pathways, hormone transport, pathogen tropism, and toxin trafficking[2,4,6]. Nevertheless, the measured efficiency left room for improvement, especially when compared to the precision-recall achieved by optimized mammalian cell CRISPR screens[23]. Here, we explored whether temporal control over Cas9 activity could achieve better resolution. We introduced a straightforward method to suppress unwanted Cas9 activity prior to integration of sgRNAs during recombination-based CRISPR screens, the IntAC approach.

In our efforts to enhance screen resolution, we introduced two key modifications at the same time: the use of the stronger *dU6:3* promoter to drive sgRNA expression and optimization of sgRNA designs using a machine learning approach based on our previous screens. By examining sgRNAs common to both the original and machine learning-based sgRNA design sets, we were able to observe a significant performance improvement due to IntAC and *dU63*, but we were unable to quantify the contribution of guide optimization per se. Formally, this would require construction of a new CRISPR library driven by *dU6:3* using the full set of v.1 sgRNAs. Due to the significant time required for new library construction and the minimal information we anticipated gaining by testing such a library, we deemed this outside of the scope of our current work. Based on work from others[23],

we do anticipate that the application of a machine learning approach can improve library design. In the future, we plan to use machine learning to evaluate sgRNA design rules based on the new, optimized v.2 screen platform data, and further optimize screen parameters based on the outcome. We also plan to test the impact of modifications to guide architecture that have proven beneficial for mammalian CRISPR screens[44].

Using IntAC along with a new *dU6:3*-driven sgRNA library enhanced resolution, resulting in the most comprehensive list of cell essential genes in *Drosophila*. Although some of these genes are likely specific to the *Drosophila* cell type studied, i.e., S2R+ cells, which are thought to have originated from hemocyte precursors[45,46], many of the genes identified in the screen are likely to reflect factors necessary for the growth and survival of all cells. Consistent with this, 90% of the previously defined 'core essential geneset 2' (CEG2) derived from mammalian screen data can be found within our *Drosophila* screen. Interestingly, in addition, comparison of the *Drosophila* cell screen data to equivalent data from human cells allowed us to predict human genes whose evolutionary duplication might mask their importance. Studies in *Drosophila* cells could help clarify the role of these crucial gene families in both invertebrates and mammals. For conserved gene families, this work underscores the utility of straightforward single-gene knockout screens in *Drosophila* cells to identify genes that would be missed in similar screens in mammalian cells.

We also show that IntAC screens display precision in positive selection screens. Using a toxin known to rely on cellular GPI anchors, PA, we show that we can identify many of the key GPI anchor synthesis pathway components (18/23) (Fig. 5A–C, Supplementary Data 1), with one additional gene (*PIG-Wa*) showing modest enrichment. As this is the first demonstration of a functional genomic screen to identify PA resistance genes in *Drosophila* to our knowledge, we cannot rule out that the missing genes are not required for killing by PA, as only one of these was previously shown to have a role in GPI anchor synthesis in *Drosophila* (*PGAP5*). Moreover, some genes necessary for GPI anchor synthesis in other contexts are largely dispensable for killing by PA in human cells (e.g. *PIG-G, -Z, PGAP1, -3, -4*), suggesting that some modifications within the pathway can be skipped and nonetheless result in a form of the GPI anchor ultimately recognized by PA[35]. Further experiments will be required to explain why these genes were missed experimentally. Overall, while IntAC screens have impressively low false negatives in genome-wide positive selection screens, there remains room for further optimization (especially of sgRNA design) to achieve even greater completeness.

In addition to GPI anchor synthesis components, we identified many components of complex N-glycan synthesis in the Golgi apparatus. Several studies have suggested the importance of complex N-glycans for susceptibility to PA. For instance, erythrocyte glycophorin, a heavily glycosylated but non-GPI-anchored protein, has been suggested to be a receptor or co-receptor[47]. Additionally, studies in Chinese Hamster Ovary (CHO) cells, using a fluorescein-conjugated inactivated aerolysin, reported that genetic ablation of the key gene that produces complex N-glycans, *Mgat1*, or chemical inhibition of the next committed step in their synthesis, α-Manosidase-II inhibition by swainosine, severely reduces binding[36]. Our analysis aligns with these studies by identifying several key genes in the pathway as essential for

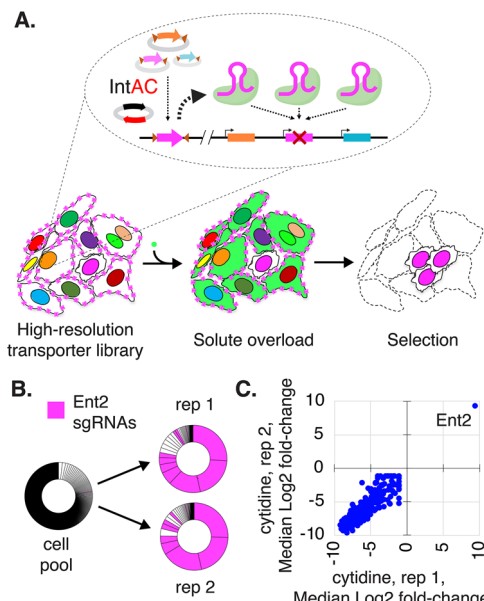

**A.**

High-resolution transporter library | Solute overload | Selection

**B.** Ent2 sgRNAs

cell pool → rep 1, rep 2

**C.** cytidine, rep 2, Median Log2 fold-change vs cytidine, rep 1, Median Log2 fold-change — Ent2

**Fig. 7 | Positive selection of a nucleoside transporter using IntAC in a solute overload screen. A** Schematic of a solute overload screen using the IntAC platform. Cells were transfected with a targeted sublibrary of sgRNAs focused on 389 predicted solute carrier genes, using the *dU6:3* promoter to drive sgRNA expression. Following transfection, the cells were subjected to solute overload, specifically with adenosine and cytidine, to identify genes involved in nucleoside transport and cellular resistance to solute toxicity. Elements created in BioRender. Viswanatha, R. (2025) https://BioRender.com/0w8vn0z. **B** Representation of sgRNA distribution following cytosine overload. All ~4000 sgRNAs in the library are represented as slices of the donut chart in descending abundance from 12:00 clockwise. Following two independent overloads with cytosine, different *Ent2* sgRNA sequences (pink) occupy ~75% of reads and the first 7 positions of replicate 1 and the first 5 positions of replicate 2. **C** Gene-level Z-scores show that *Ent2* is the only gene detected in both replicates.

PA killing, including *Mgat1* and *α-Manosidase-II*, themselves (Fig. 5B). Interestingly, the only prior genome-scale screen for PA resistance (conducted in human HAP1 cells) failed to identify N-glycan modifiers, retrieving only GPI anchor synthesis genes[34]. One possible explanation is that the requirement for complex N-glycans in PA binding and subsequent homoheptamerization to form a membrane pore, resulting in cell killing, reflects the distribution in the type, density, and geometry of GPI-anchored proteins displayed on the surface, a property that varies from cell type to cell type.

In our PA resistance CRISPR screen, we also identified a *Drosophila* gene regulating the well-studied GPI anchor synthesis pathway, *CG46311*. *CG46311* is annotated as a small open reading frame (smORF) gene (40 amino acids) within a larger gene, *Tpst*. smORFs, defined as ORFs encoding a polypeptide of less than 100 amino acids, typically regulate the larger downstream ORF, and were historically overlooked when compiling genome annotations[48]. *CG46311* was not added until FlyBase version 6, so it was not included in our v.1 CRISPR screening library, which was based on FlyBase version 5. *CG46311* sgRNAs were enriched in the v.2 IntAC screen, whereas none of the 7 sgRNAs targeting *Tpst* were enriched. Furthermore, we could rescue the *CG46311* knockout phenotype (failure to display a GPI-anchored protein on the cell surface) by overexpressing the *CG46311* coding sequence alone, strongly suggesting that CG46311 loss but not Tpst loss is responsible for the GPI anchor defect. Moreover, Alphafold modeling predicts that CG46311 binds tightly in the *Drosophila* GnT complex and adopts a similar position within the complex as that of PIG-Y, a core GPI anchor synthesis component thought to be missing from *Drosophila*. We therefore propose that *CG46311* represents the *PIG-Y* ortholog and constitutes a core GPI anchor synthesis protein.

We additionally developed a focused CRISPR library targeting *Drosophila* solute transporters, enabling the identification of transporters responsible for solute overload using smaller cell populations than are needed to conduct gnome-wide screens. This will enable the creation of comprehensive solute-to-transporter maps and provide a platform for in vivo follow-up studies in *Drosophila*. Such insights can be crucial for understanding transport mechanisms, including homologous processes in human cells.

Given the robust performance of the IntAC approach in *Drosophila* cells, the next goal is to apply the approach in other cell species. This could have a particular impact on non-model species for which cell lines exist, but the application of large-scale genetic screening would be novel. This includes cell lines derived from lepidopteran insect pests and arthropod vectors of disease, such as mosquitoes and ticks. Given the lack of any functional genomic data, genome-wide screens enhanced by IntAC could be used to construct improved gene annotations and reveal important evolutionary insights of cell-essential genes. Furthermore, the identification of essential genes in mosquitoes and ticks can be used as targets for gene drives designed for population suppression[49,50]. As we attempt to apply the IntAC approach in additional cell lines, there is some potential for further challenges. The IntAC method relies on the loss of the anti-CRISPR encoding plasmids over time; efficient random integration of the plasmid in a given cell type could lead to the presence of a large population of cells that have integrated sgRNAs but permanently inactivated Cas9, reducing the genotype–phenotype linkage. Therefore, the balance between the site-specific versus random integration in new systems will have to be tested and optimized. Second, the method relies on the availability of efficient anti-CRISPRs for the Cas enzyme utilized. Thus, IntAC is amenable in principle to all screens using Cas9 or Cas12a (inhibited by the AcrVA family[51]), including kinase-dead or nicking variants for CRISPR activation or inhibition[1], prime editing[52], or base editing[53], whereas there is currently no anti-CRISPR for the RNA-cleaving Cas13d/CasRx. Thus, while the method is widely applicable to many currently used screening types, the universality will depend on the development of precise anti-CRISPRs for each Cas. Finally, IntAC could also potentially enhance mammalian virus-free CRISPR screens, which have similarly used a plasmid transfection-based attP-attB recombination to deliver sgRNAs followed by a second transfection to deliver Cas9. These have been demonstrated in CHO, HEK293, and K562 cells[14,15,54–56], where the key motivations were to design CRISPR screens that expressed a constant dosage of sgRNAs from a defined locus and to avoid biosafety constraints of using viral vectors. In such cases, IntAC could offer a promising addition by enabling temporal control of Cas9 activity in a single transfection step and lead to improved screening resolution.

## Methods

### Plasmids
pLib6.6 (Addgene # 176652) was constructed by replacing the *dU6:2* promoter from pLib6.4 (Addgene #133783) with the *dU6:3* promoter from pCFD3 (Addgene #49410[20]). A synthetic gene fragment containing the anti-CRISPR AcrIIa4 codon-optimized for *Drosophila* was attached downstream of the φC31[2] integrase (from pBS130, Addgene #26290), followed by a P2A site. The resulting fusion was cloned downstream of the *Drosophila* Actin promoter in pAWF using standard cloning methods to generate pIntAC (which will be made available on Addgene shortly). As a control, pInt was generated for this study by cloning φC31 integrase (from pBS130, Addgene #26290) into pAWF. pAc5-CD58^GPI-GFP, a tracer along the GPI anchor pathway, was constructed by subcloning the ORF from Addgene #182866 into pAc5-STABLE-neo (Addgene #32425). *CG46311* and *CG46311_Long* expression vectors were constructed by GenScript. The synthetic codon-optimized CDS, followed by a stop codon (untagged) for each gene, was subcloned into pAc5.1/V5-His B (Thermo). To prepare vhhGFP4-

ALFA-HIS, vhhGFP4 was amplified from pUASTattB_NSlmb-vhhGFP4 (Addgene #35577[57]) and subcloned into pET-26b-Nb-GGA f to construct pET-26b-vhhGFP-AH. The anti-GFP nanobody was purified using a previously reported protocol[58].

## Cell lines and treatments

The *Drosophila* S2R+ derivative PT5 (NPT005; DGRC #229) was transfected with pMK33/Cas9[2] or with pDmAct5C::Cas9-2A-Neo[3] and maintained in either 200 ng/μL Hygromycin B (Calbiochem) or 500 μg/mL G-418 (Goldbio). The efficiency of CRISPR sgRNAs in each cell line was identical, and the cell lines were used interchangeably for these experiments. Proaerolysin was a generous gift from Dr. S. Peter Howard, College of Medicine, University of Saskatchewan. Cytidine (Sigma, C122106) was dissolved in Schneider's media to prepare a 2x stock to achieve a final concentration of 250 μM. CG46311 knockout S2R+ cells express CG46311 sgRNA-1 (5′- AGC CGA AAT GCC AAA TAG AC-3′). PIG-A knockout S2R+ cells were prepared using a previously described homology-directed repeat knockout method using four sgRNAS: sgRNA-39494 (5′-TTT ACC GAT CAC AGC CTC TT-3′), sgRNA-39492 (5′-CCA TCT GTG TAT CTC ACA TA-3′), sgRNA-39495 (5′-TTT GCG GTA GAC AAG ACG GG-3′), and sgRNA-39496 (5′-GAG CGG TGC TAC TTC GTG AG-3′)[59].

## Microscopy

Live-cell, widefield images were acquired on an Evos 5000 Inverted Microscope equipped with a ×20 objective and GFP and Texas Red filters. For immunofluorescent detection of cell surface GFP, cells were fixed for 15 min in 4% paraformaldehyde, washed extensively in PBS, and then treated with a GFP nanobody, vhhGFP4-ALFA-HIS (2 ng/mL), for 1 h at room temperature. Subsequently, cells were washed, and then treated with a secondary antibody, FluoTag®-X2 anti-ALFA, Alexa Fluor 647 (NanoTag Biotechnologies), including Hoescht (Sigma), for 1 h at room temperature. Finally, cells were washed extensively in PBS and imaged by confocal microscopy. Confocal images were acquired on a Nikon Ti inverted microscope equipped with a W1 Yokogawa Spinning disk with 50 μm pinhole. In all cases, cells were seeded on Perkin Elmer 384-well clear-bottom dishes.

## T7 Endonuclease-I assay

A pLib6.6 plasmid expressing an sgRNA targeting Rho1 was used alongside a plasmid encoding φC31 integrase (Int) or φC31 integrase-2A-ActIIa4 (IntAC) under the *Drosophila* Actin promoter, and cells were maintained in puromycin selection starting from 4 days following transfection. The *Rho1* target genomic region was amplified by PCR using specific primers flanking the CRISPR/Cas9 editing site (m-Rho1-seq20_0F, 5′-GGT GCC TGC GGT AAA ACT TG-3′ and dm-Rho1-seq20_0R, 5′-ATC ACT TGG ATG GCA GGG TG-3′). The PCR products were gel extracted and then denatured at 95 °C for 5 min and re-annealed by gradually cooling to room temperature to form hetero-duplex DNA. The re-annealed DNA was then digested with 10 units of T7 Endonuclease I (New England Biolabs, Cat# M0302) in 1X NEBuffer 2 at 37 °C for 30 min. The reaction was stopped by adding 0.25 M EDTA, and the products were analyzed on a 2% agarose gel stained with SYBR Safe DNA stain (Thermo). Band intensity quantification was performed using ImageJ.

## CRISPR library construction and delivery

For the v.2 library, CRISPR sgRNAs were designed independently using the following approach. First, all putative CRISPR sgRNAs were retrieved from the Drosophila genome (Flybase version 6.24). Next, variant calling was used to remove single-nucleotide polymorphisms within CRISPR sgRNAs. Among retained sgRNAs, priority was given to those with a high machine learning (ML) score. The pipeline for sgRNA selection is available at [https://www.flyrnai.org/crispr3/web/]. The CRISPR library was synthesized as an oligo pool between constant

sequences (5′ sequence: TAT ATA GAC CTA TTT TCA ATT TAA CGT CG; 3′ sequence: GTT TTA GAG CTA GAA ATA GCA AGT TAA AAT) (Genscript), amplified with outside primers corresponding to the 5′ constant sequence and the reverse complement of the 3′ constant sequence in 17 cycles using Phusion Polymerase (New England Biolabs), and then inserted into BbsI-digested pLib6.6 using the NEB HiFi Assembly Master Mix Library. The reaction was electroporated into E. cloni 10 G ELITE cells (Lucigen) and plated on 150-mm LB-Carbenicillin selective plates and grown overnight at 30 °C. This procedure was scaled such that the number of resulting colonies was at least 100 times the library complexity. The library is available on Addgene (#239984). Transfection in *Drosophila* S2R+ derivative cell lines was performed as previously described[2] using Effectene reagent along with minor adjustments for cost savings (detailed in Supplementary Fig. 2A). All libraries were transformed with pInt or pIntAC at a molar ratio of 1:1. Briefly, 5 μg of library and 5 μg of either pInt or pIntAC were mixed with 80 μL of Enhancer Solution, 1500 μL of EC buffer, and 300 μL of Effectene (Qiagen). The transfection mix was added to 50 mL of actively growing S2R+ cells adjusted to 2.4 × 10⁶ cells/mL, mixed, and 5 mL distributed to each of ten 100-mm dishes and shaken to distribute the cells evenly. Plates were tightly sealed in plastic film and incubated overnight at room temperature. The next day, 5 mL of fresh media was added to each plate to prevent evaporation. After three additional days, cells were selected in puromycin-containing media. They were then grown for 3 additional weeks under puromycin selection with media changes every 4–5 days before being used for subsequent experiments. For chemical selection experiments, chemicals were added at this point. PA was added at 2 nM for 3 weeks. Cytidine was added at 250 nM for 4 weeks. For fitness gene assessment, cells were passaged for an additional 4 weeks using the same regimen.

## Library sequencing and CRISPR screen data analysis

The sgRNA counts were prepared using a genomic DNA PCR protocol established previously[2], with minor modifications. First, cells were lifted from plates and resuspended at a concentration of 5–20 × 10⁶ per mL, and 25–50 mL of cells was pelleted in a 50 mL conical tube and the media discarded. Pellets were stored frozen. Next, eight Zymo gDNA miniprep (D3025) columns were used per frozen pellet using the manufacturer's instructions, resulting in the isolation of 300–400 μL of gDNA per sample. Samples were diluted 1:1 with water and again 1:1 with 2X GoTaq Master Mix (Promega M712B) containing appropriate primers and amplified in 23 PCR cycles. Samples were resolved on a 1% agarose gel, and the ~450 bp band corresponding to the sgRNA expression cassette containing a mixture of sgRNA sequences was gel-purified. Next, outside primers adding P5 and P7 sequences were used to reamplify the product using Phusion Polymerase (New England Biolabs). The concentration of DNA in each sample was next quantified using a Cubit fluorometry assay (dsDNA, Broadrange, Thermo) and then normalized and mixed together. Combined samples were then directly loaded on a Novaseq 6000 lane (Biopolymers Facility, Harvard Medical School), and sequencing was conducted with the goal of sequencing each sgRNA 100 times or more. All data processing used MAGeCK version 0.5.6.

## Barcoding strategy

We used a two-step fingerprinting-and-barcoding strategy similar to that previously employed[2]. A portion of the Illumina Read1 primer, unique experimental fingerprints (varying stretches of random nucleotides), and 6-nucleotide barcodes were added to the first PCR round (PCR1). The second PCR (PCR2) then anneals to the PCR1 product and adds extensions containing the 5′ P5 site and 3′ P7 site to enable sequencing on Illumina sequencers. An example of one forward primer used for the first PCR step is: 5′-CCT ACA CGA CGC TCT TCC GAT CTN NNN TTG GCT CCT ATT TTC AAT TTA ACG TCG-3′, where "NNNN" represents the fingerprint and "TTGGCT" represents the

barcode. "CCT ATT TTC AAT TTA ACG TCG" aligns to the dU6:3 primer and extends the unique sgRNA spacers. The common PCR1 reverse primer is 5′-TTT GTG TTT TTA GAA TAT AGA ATT GCA TGC-3′. The common PCR2 forward primer (adding P5) is: 5′-AAT GAT ACG GCG ACC ACC GAG ATC TAC ACT CTT TCC CTA CAC GAC GCT CTT CCG ATC T-3′. The common PCR2 reverse primer (adding P7) is: 5′-CAA GCA GAA GAC GGC ATA CGA GAT TTT GTG TTT TTA GAA TAT AGA ATT GCA TGC-3′.

## Bioinformatics

CRISPR screen analysis used MAGeCK version 0.5.4 or 0.5.6 running on the Harvard Medical School O2 cluster. Graphics were prepared with help from Biorender. Graphs were prepared using ggplot2 in R Studio (version 2023.12.1+402) or Microsoft Excel (version 16.89.1). Updated genome-wide libraries were prepared using homemade code using data from FlyBase version 6. Information about genes was gathered from FlyBase version 6. DIOPT version 7 was used for predicting orthologs. For GnT complex prediction, Alphafold-Multimer[60] (Alphafold3) was provided with *Drosophila* GnT protein sequences (PIG-A, isoform B: NP_001286547.1, PIG-C, isoform B: NP_647804.2; PIG-P: NP_651459.2; PIG-H, isoform B: NP_001262358.1; PIG-Q, isoform C: NP_001285323.1; Dpm2, NP_001163338.1; CG46311: NP_001334677.1) or human GnT protein sequences (PIG-A, isoform 1: NP_002632.1; PIG-C: NP_002633.1; PIG-P, isoform 1: NP_710148.1; PIG-H, isoform 1: NP_004560.1; PIG-Q, isoform 1: NP_683721.1; DPM2, isoform 1: NP_003854.1; PIG-Y: NP_001036081.1) and ribbon structure as well as interface view was captured from viewing model 0 in Chimera X (version 1.9). For visualization of PAE and ipTM plots, all models were passed through the Local Interaction Score Analysis pipeline (Github flyark/AFM-LIS)[61].

## Statistics and reproducibility

No statistical method was used to predetermine sample size. No data were excluded from the analyses. The experiments were not randomized. The Investigators were not blinded to allocation during experiments and outcome assessment.

## Reporting summary

Further information on research design is available in the Nature Portfolio Reporting Summary linked to this article.

## Data availability

Custom code used to pick sgRNAs is available on Github under the MIT license [https://gitlab.com/hms8653705/pick-crispr]. A web-based sgRNA search tool (allowing batch search queries) that displays sgRNA sequences along with parameters used to prioritize them in screening libraries is available [https://www.flyrnai.org/crispr3/web/]. Genome-wide v.2 library is available through Addgene (#239984). All other plasmids generated for this study are available upon request. Unprocessed sequencing files (FASTQ) are available on SRA. Accession code PRJNA1263494. Source data are provided with this paper.

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

## Acknowledgements

We thank Dr. Baolong Xia, Ahmad Muhammad, and Dr. Pratyajit Mohapatra for testing the method. We thank Dr. Adam N. Carte for constructing the GFP-HsCD58^GPI vector. We thank Dr. S. Peter Howard for his generous support in providing proaerolysin (PA). We additionally thank the Microscopy Resources on the North Quad (MicRoN) core at Harvard Medical School for imaging support. This work was supported by NIH P41 GM132087 to S.E.M. and N.P as well as a grant from the Merkin Institute for Transformative Technologies to N.P. In addition, N.P. is a Howard Hughes Medical Institute (HHMI) investigator. This article is subject to HHMI's Open Access to Publications policy. HHMI lab heads have previously granted a non-exclusive CC BY 4.0 license to the public and a sublicensable license to HHMI in their research articles. Pursuant to those licenses, the author-accepted manuscript of this article can be made freely available under a CC BY 4.0 license immediately upon publication.

## Author contributions

R.V. conceived of the IntAC approach and conducted the genome-wide screens and follow-up experiments. S.E. conceived of the solute overload approach and performed the cytidine overload screen. Y.H. and R.V. performed bioinformatics for sgRNA design, ortholog mapping, and CRISPR screen data analysis. K.R., R.V., and M.B. amplified and sequenced sgRNAs from genomic DNA. A.-R.K. prepared the GFP nanobody and helped with Alphafold modeling. M.Q. reanalyzed RNA-seq data. S.E.M. and N.P. supervised and provided instrumentation and reagents. R.V., S.E.M., and N.P. wrote the manuscript.

## Competing interests

The authors declare no competing interests.
