## [Transparent Peer Review file · Nature Communications]

Higher resolution pooled genome-wide CRISPR knockout screening in *Drosophila* cells using integration and anti-CRISPR (IntAC)

Corresponding Author: Professor Norbert Perrimon

Version 0:

Reviewer comments:

Reviewer #1

(Remarks to the Author)

Pooled CRISPR screens have revolutionized functional genomics in mammalian cells by enabling systematic, high-throughput investigation of gene function. Their power lies in the ability to introduce single guide RNAs (sgRNAs) into large populations of cells, typically through lentiviral vectors at low multiplicity of infection (MOI), ensuring that each cell receives only one unique sgRNA. This one-to-one relationship between cells and sgRNAs is crucial for accurate phenotype-to-genotype associations and minimizes confounding effects from multiple genetic perturbations. By leveraging this approach it has been possible to map essential genes in various cell types and identify genes essential for drug responses or disease mechanisms. However, similar approaches have been limited in non-mammalian models, mainly due to a lack of methodology to introduce single sgRNA into cells in a pooled format.

The authors of this manuscript have previously published a system for pooled CRISPR screening in insect cells based on the single copy integration of sgRNA plasmids using the PhiC31/attP/attB system. In the current manuscript they report significant improvements to this system. These are based on inhibiting Cas9 activity until the integrated transgene is the only source of sgRNA in the cell. They demonstrate that this results in a marked increase in sensitivity and resolution in pooled fitness screens (drop-out screening), with now similar results to established mammalian screens, and demonstrate utility in positive selection screens and report some interesting biological discoveries along the way.

The study is well designed and the experiments support the conclusions drawn. The manuscript is also very well written, although the presentation of some of the figures could be improved. Overall, I find this to be an important advance in the field that will improve functional genomics in *Drosophila* cells. The authors suggest that this technology could also be extended to other insect systems (e.g. disease vectors, pest species), although no steps in this direction are presented.

Below are some questions and suggestion with the aim to help the authors to improve the manuscript further:

In the PA screen 5/23 (~20%) of expected genes were not identified. This is a substantial number and it would be important to know if this reflects differences in human vs *Dmel* cell biology or technical limitations of the screening setup. To this end it would be informative to employ the sgRNAs encoded in the library that target expected hits, but fail to enrich in resistant cells, and use them in an arrayed format. This way the authors can test if the sgRNA successfully edit the target gene or not and if yes, if mutant cells acquire resistance in this experimental setup.

The set of essential genes identified in this study should be made more easily accessible. The authors provide raw data in Suppl. Table 1, but this is not very easy to parse. Additional lists with genes classified as essential at 0.1%, 1% and 5% FDR should be provided.

What genome was used to design the sgRNA library? In our experience, discrepancies between the experimental genome and the one used to design sgRNAs is a major driver of inactive sgRNAs. Generally, the material and methods section should explain how the sgRNA library was designed.

The authors compare the efficiency of discovering essential genes with CRISPR v2 with that of v1 and fitness screens in

human cells. It would be useful to also compare the efficiency of v2 to past arrayed RNAi screens. Many scientists have performed arrayed RNAi screens in S2 cells over the last two decades and will ask themselves if it would be worthwhile to revisit some of these screens with v2 CRISPR.

CG46311 appears to be an upstream open reading frame of the *Tpst* gene. As such it is expected to regulate translation efficiency of the downstream ORF and mutagenesis is therefore likely to also affect this gene. This makes the rescue experiments particularly important and the authors might want to point this out.

Page 8: "Knockout cells displayed a nearly complete lack of membrane-targeted GFP-HsCD58GPI, a canonical tracer of the pathway [38]." This sentence lacks a call out to Fig. 5f. In addition, this figure is not very convincing, as it is difficult to distinguish between membrane-targeted and intracellular GFP. Is it possible to perform an antibody staining in the absence of detergent to selectively visualize the protein pool that reaches the plasma membrane?

Figure 1a and b could be improved. For example, one difference between v1 and v2 appears to be that the transient plasmids create the orange and purple sgRNAs in v1, but orange, purple and blue in v2. There are also more sgRNA in v2, likely a reference to U6:3, but this is not indicated in the figure. On the right side of the figure mutations are indicated by a scissor, which is not ideal (in particular for the orange gene), as these are not actively cut at this stage. It would also be good to indicate that the important result of the "cell divisions" is the loss of the non-integrated plasmids.

Figures 5a and c are quite confusing and insufficiently labeled and explained. For example, what are the green, yellow, purple and orange boxes, triangles and ovals?

Results page 5. "Moreover, this increase was not observed when the anti-CRISPR protein was left out (v.2 library with ϕ C31 integrase alone, 'Int') (Figure 3A)." Fig.3A also nicely shows how sgRNA targeting non-expressed genes are associated with strong effects in this setup. You make this point in the figure legend, but I would also mention it in the main text, as it is important and illustrates why inhibition is so important.

Page 5: "These results suggest that much of the improvement between v.1 and v.2 is driven by higher sgRNA expression levels via the dU6:3 promoter as made possible by use of the IntAC approach". I guess another important factor is that there is a better 1-to-1 relationship between detected sgRNA and edited gene(s) in v.2.

Introduction: "Tighter strategies for drug-controlled expression such as 'tet-on' are lacking in *Drosophila*." - Tet-on is also notoriously leaky in mammalian cells when combined with Cas9, so you might want to rephrase or remove that sentence.

Typo in Figure 2b: taRgeting

Suppl. Fig. 2a: Indicate on the y axis that these are consecutive weeks.

Reviewer #2

(Remarks to the Author)

General comments

This work from Viswanatha and colleagues is aimed to improve the effectiveness of CRISPR induced knockout screening in cultured S2R-PT5 (*Drosophila*) cells. The authors provide a modified version of a previously published system "Pooled genome-wide CRISPR screening for basal and context specific fitness gene essentiality in *Drosophila* cells eLife, 2018" based on site-specific recombination of sgRNA pools in Cas9-expressing cells as alternative to methods relying retroviral vector delivery.

The effectiveness of the previous approach was heavily relying on the use of "weak" promoters to mitigate the effect of "early multigenic indels" that caused discrepancies between the integrated sgRNA and related genetic/phenotypic outcomes and cell lethality. However, the use of weaker promoters is not always ideal and can reduce the effectiveness of such tools. To mitigate the unwanted early editing the authors add a transient dose of an anti-CRISPR protein (namely AcrIIA4), which is a well-known inhibitor of Cas9/gRNA DNA binding activity.

The authors provide a comprehensive set of data to show that this addition substantially improves the power of the system enabling to identifying a more comprehensive set of "fitness/essential" genes in *Drosophila* cells as well applicable to the identification of genes linked to specific phenotypes such as proaerolysin resistance and solute transport. Although none of the components or steps provided are exceptionally innovative or the mitigating effect brought about by the addition of the anti-CRISPR are particularly surprising, I believe that the work provides a substantial improvement of CRISPR-based screening in cultured cells with the due general limitations that invitro system may have vs invivo functional studies.

I have enjoyed reading the manuscript and have not noted any major concerns, the authors are clearly well familiar with the methodology and provide a sound analysis of results as support of the key claims. The authors also provide a fair appraisal of advantages and limitations of current and foreseeable applications of the system in the discussion. Here are a few minor comments that I have noted whilst reading the manuscript.

Specific comments

Abstract

"IntAC dramatically improves precision-recall of fitness genes across the genome, allowing us to generate the most comprehensive list of cell fitness genes yet assembled for *Drosophila*."

"Fitness genes" might not mean much to many particularly in a cell-specific invitro context, rather than actual living insects,

which should probably be specified or changed to “essential” for *Drosophila* cells?

Results

“AcrIIa4 is the most potent known protein inhibitor of Cas9 activity [17-19]. This protein mimics guide RNAs, binding and functionally inactivating Cas9. Cas9 activity is thereby...” My understanding is that Acr proteins mimic DNA binding rather than gRNA, which I believe is still able to form the RNP complex. The latter is unable to interact with the DNA target due to the presence of the Acr. Please doublecheck and eventually revise accordingly.

“This finding supports our hypothesis and pilot experiments suggesting that early, uncontrolled Cas9 activity in the absence of anti-CRISPR reduces the precision of the screen, particularly when combined with elevated sgRNA level due to our use of the stronger dU6:3 promoter.” Is this assumed from the wider positive z-score shown in 3A top right plot? It might be useful to expand a little more to better explain this.

Discussion

“This work underscores the utility of straightforward single-gene knockout screens in *Drosophila* cells to identify genes that would be missed in similar screens in mammalian cells.” I assume this would only be applicable to conserved genes/phenotypes. I would suggest specifying the caveat.

Figure 1A/B. I can't see a description of the plasmid indicated as “plnt” in the figure

Figure 4B and related text “1 to many” is not very helpful to indicate if a specific N of orthologues threshold is applied or any gene >1 orthologues is considered as “many”. Please change/specify accordingly

Reviewer #3

(Remarks to the Author)

Viswanatha and colleagues present a novel whole-genome CRISPR screening platform in *Drosophila melanogaster* cell lines, which has the potential to be adaptable to other invertebrate species such as important agricultural pests and disease vectors. The new approach described is an improvement on a previous whole-genome CRISPR screening approach developed by the same group. The main advancement is the inclusion of the anti-CRISPR AcrIIa4, a Cas9 inhibitor that prevents gene editing at early times after transfection and thus reduces gene editing prior to integration of gRNAs that are later identified by gene sequencing, resulting in a higher hit rate and reduced false-discovery rate. The authors present an in-depth validation of their new screening platform and directly compare it to their previous approach to clearly demonstrate the improvements made. The application of whole genome CRISPR screens in invertebrates has lagged behind their use in human cells, largely because lentiviral transduction is not possible. The data presented in this manuscript therefore represent a major technological advance that will likely stimulate additional important functional screens in *Drosophila* and other non-model organisms.

In the present manuscript, the authors additionally use their screening platform to determine the first high-accuracy list of essential genes in *Drosophila*, using this to demonstrate that gene duplications in the human genome limit the ability of whole genome CRISPR screens to identify essential genes in humans. The authors also confirm the utility of their new approach in a functional screen that identified a novel regulator of GPI-anchor protein trafficking, and in a targeted screen for nucleoside transporters.

Overall, the work represents a major advancement in the toolkit available for studying genotype-phenotype relationships in invertebrates that will have a broad and major impact beyond the immediate functional questions and organisms investigated in this manuscript.

Major comments:

- (1) Do the authors have any data to confirm that only one gRNA becomes integrated into each attP site/cell? This would be important to properly interpret the specificity and sensitivity data presented later in the paper.
- (2) In Fig 1, the authors do not show that editing in their new system reaches a maximum 18 days post-transfection, as this is the latest time point tested. They would need to test additional (later) time points to make this claim.
- (3) The authors state in the Results section that their AI-improved gRNA design had only a minor impact on the improvements to their screen. However, as they acknowledge in the Discussion, they do not actually have any evidence to support this claim. An additional screen, potentially with a subset of genes, would be required to support this claim.
- (4) In Fig 5, the authors note that several genes essential for GPI anchor protein synthesis and complex N-glycosylation pathways were not picked up in their PA resistance screen. To more firmly verify the specificity of their improved screen, the authors would need to transiently silence (or knock out) the three genes that have not previously been investigated for a role in these pathways.
- (5) CG46311 is identified as a previously unidentified orthologue of human PIG-Y (previously thought not to have an orthologue in *Drosophila*). These findings would be more convincing if sequence alignments and structural similarity comparisons of the AlphaFold structural predictions of PIG-Y and CG46311 were presented.
- (6) The microscopy images in Fig 1F are difficult to interpret. Confocal images and quantification should be shown to strengthen these data as they are essential for confirming the function of the putative novel PIG-Y orthologue.

Minor comment:

- (1) A number of terms used may not be clear to the broad audience for which this manuscript will be of interest. Please define "isotopically overgrown", "dropout" and "passengers" to improve the clarity of the manuscript.
- (2) Can the authors explain how Rho1 appears in their list of essential genes when they were able to target this gene in their validation experiments in Fig 1D?
- (3) In Fig 1, a key or more clear description in the figure legend would be helpful to make Fig 1A and B easier to interpret.
- (4) In Fig 1D, what do the green/red colours represent in the microscopy images?
- (5) I found the inclusion of the AlphaFold structural predictions in Fig 5F distracting and unnecessary.

Version 1:

Reviewer comments:

Reviewer #1

(Remarks to the Author)

The authors have significantly improved the manuscript and have addressed my main concerns adequately. I wish to congratulate them on this nice body of work and look forward to seeing the discoveries this technology will enable in the years to come.

Minor point:

Typo in Line 307: GFP-GPI was UNdetectable on the surface of CG46311 KO cells. In the respective figure there is also a typo in the label ("CG4311").

Reviewer #2

(Remarks to the Author)

The authors have addressed all my previous concerns and I do not have other comments.

Reviewer #3

(Remarks to the Author)

The authors have provided extensive additional experimental data and analysis, as well as rewording parts of the manuscript, and my previously raised concerns have all been addressed.

Response to reviewers (NCOMMS-24-64911)

Here we present our revision of “**Higher resolution pooled genome-wide CRISPR knockout screening in Drosophila cells using integration and anti-CRISPR (IntAC)**” for your consideration. We thank the reviewers for their time and detailed comments, and we thank you all for providing us the time to address them.

In addition to addressing the reviewer’s points, we also identified an error in the manuscript: a cloning mistake led to the incorrect assertion that a long isoform and not the annotated CG46311 smORF rescues the CG46311 knockout. After re-sequencing the rescue constructs, we found that both the smORF as annotated as well as a putative longer isoform completely rescued the defect. We additionally reviewed RNA sequencing data and agree with FlyBase’s assertion about the most likely annotation of CG46311. This mistake does not alter the conclusions of the paper or address specific reviewer comments. We are extremely sorry for the error and glad we identified it in time. We revised the manuscript text and figures to account for it.

Below, please find our detailed answers to reviewer questions.

Reviewer 1

1. *In the PA screen 5/23 (~20%) of expected genes were not identified. This is a substantial number and it would be important to know if this reflects differences in human vs Dmel cell biology or technical limitations of the screening setup. To this end it would be informative to employ the sgRNAs encoded in the library that target expected hits, but fail to enrich in resistant cells, and use them in an arrayed format. This way the authors can test if the sgRNA successfully edit the target gene or not and if yes, if mutant cells acquire resistance in this experimental setup.*

We agree that exploring the contribution of the five genes not retrieved in the screen is important, but it lies beyond the scope of this study. Our objective here was to identify the major GPI anchor synthesis genes necessary for PA resistance, rather than to verify every predicted hit. Moreover, we now apply the “homology-directed insertion” approach to successfully generate a PIGA mutant that displays a lack of surface GPI anchors (Supplementary Figure 5C), which illustrates an approach that can be extended to individually knock out these five genes in future work. Thus, we believe that detailed functional studies of these candidates are outside the aims of this manuscript.

2. *The set of essential genes identified in this study should be made more easily accessible. The authors provide raw data in Suppl. Table 1, but this is not very easy to parse. Additional lists with genes classified as essential at 0.1%, 1% and 5% FDR should be provided.*

We agree with the reviewer and now provide a separate tab within the supplementary table that contains the genes identified at 0.1%, 1%, 2%, and 5% FDR.

3. *What genome was used to design the sgRNA library? In our experience, discrepancies between the experimental genome and the one used to design sgRNAs is a major driver of inactive sgRNAs. Generally, the material and methods section should explain how the sgRNA library was designed.*

We thank the author for this thought! To minimize the effect of inactive sgRNAs, we removed those with SNPs with the *Drosophila* S2R+ genome. We apologize for the oversight and have now added text to the Materials and Methods section to include the version of the *Drosophila* genome used (6.24). We note that the sequences of the library are provided in Supplementary Table 1. Now, we add a note to the Materials and Methods that users can access our sgRNA database showing all parameters (including whether or not there are SNPs with the S2R+ genome sgRNA sequences) at <https://www.flyrnai.org/crispr3/web/>.

4. *The authors compare the efficiency of discovering essential genes with CRISPR v2 with that of v1 and fitness screens in human cells. It would be useful to also compare the efficiency of v2 to past arrayed RNAi screens. Many scientists have performed arrayed RNAi screens in S2 cells over the last two decades and will ask themselves if it would be worthwhile to revisit some of these screens with v2 CRISPR.*

While we previously showed this comparison (Viswanatha et al., 2018 eLife), we agree that this is an important point. We now add the genome-wide RNAi comparison to Figure 3A and 3C. To address some confusion in the “Z-score vs. expression plots” we now bin the genes by their predicted importance for fitness using the various approaches (genome-wide RNAi, CRISPR screen v.1, CRISPR screen v.2 with anti-CRISPR, CRISPR screen v.2 without anti-CRISPR) and display the distribution of expression levels for genes within each bin.

5. *CG46311 appears to be an upstream open reading frame of the *Tpst* gene. As such it is expected to regulate translation efficiency of the downstream ORF and mutagenesis is therefore likely to also affect this gene. This makes the rescue experiments particularly important and the authors might want to point this out.*

We thank the reviewer for his comments. We have now added a more complete discussion of CG46311 to the Discussion.

In our PA resistance CRISPR screen, we also identified a new *Drosophila* gene regulating the well-studied GPI anchor synthesis pathway, CG46311. CG46311 is annotated as a small open reading frame (smORF) gene (40 amino acids) within a larger gene, *Tpst*. smORFs, defined as ORFs encoding a polypeptide of less than 100 amino acids, typically regulate the larger downstream ORF, and were historically overlooked when compiling genome annotations [52]. CG46311 was not added until FlyBase version 6, so it was not included in our v.1 CRISPR screening library, which was based on FlyBase version 5. CG46311 sgRNAs were enriched in the v.2 IntAC screen, whereas none of the 7 sgRNAs targeting *Tpst* was enriched. Furthermore, we could rescue the CG46311 knockout phenotype (failure to display

a GPI-anchored protein on the cell surface) by overexpressing the CG46311 coding sequence alone, strongly suggesting that CG46311 loss but not *Tpst* loss is responsible for the GPI anchor defect. Moreover, Alphafold modeling predicts that CG46311 binds tightly in the *Drosophila* GnT complex and adopts a similar position within the complex as that of PIG-Y, a core GPI anchor synthesis component thought to be missing from *Drosophila*. We therefore propose that CG46311 represents the PIG-Y ortholog and constitutes a core GPI anchor synthesis protein.

6. Page 8: “Knockout cells displayed a nearly complete lack of membrane-targeted GFP-*HsCD58GPI*, a canonical tracer of the pathway [38].” This sentence lacks a call out to Fig. 5f. In addition, this figure is not very convincing, as it is difficult to distinguish between membrane-targeted and intracellular GFP. Is it possible to perform an antibody staining in the absence of detergent to selectively visualize the protein pool that reaches the plasma membrane?

We thank the reviewer for pointing out this oversight and suggesting a critical experiment and have now revised this section. As the reviewer suggests, we have performed staining after fixation without permeabilization and then detected GFP using a GFP-targeting nanobody. The results confirmed our prediction that CG46311 and not *Tpst* is responsible for the GPI defect.

In the course of these experiments, we detected a cloning error in our CG46311 rescue construct. We determined that the smORF, as annotated by FlyBase, is capable of fully rescuing the GPI anchor defect, to the same extent as our longer ORF. Moreover, we reanalyzed expression data and confirmed that most detected transcripts do, indeed, match the predicted ORF (Supplementary Figure 5B). We apologize for this error, but note that it does not change the key observation that *CG46311* and not *Tpst* is responsible for the defect. We made several changes to the text and Discussion section to account for this error.

7. Figure 1a and b could be improved. For example, one difference between v1 and v2 appears to be that the transient plasmids create the orange and purple sgRNAs in v1, but orange, purple and blue in v2. There are also more sgRNA in v2, likely a reference to *U6:3*, but this is not indicated in the figure. On the right side of the figure mutations are indicated by a scissor, which is not ideal (in particular for the orange gene), as these are not actively cut at this stage. It would also be good to indicate that the important result of the “cell divisions” is the loss of the non-integrated plasmids.

We thank the reviewer for providing excellent suggestions to convey important information in this complicated diagram. We now add “loss of transiently transfected plasmids” to the diagram for clarity. And remove the scissors from the left-hand side of panel A. Furthermore, we now provide a detailed description of the diagram in the figure legend.

Figure 1: Schematic of the IntAC system and validation of transient inhibition by anti-CRISPR in *Drosophila* cells. (A,B) Schematic representation of the current

CRISPR screening system compared with the IntAC system. (A) In the v.1 approach, sgRNAs are active upon pooled transfection and one of these at random integrates into the cell's genome via ϕ C31 integration. Following cell divisions and the loss of transiently transfected plasmids, one sgRNA, illustrated by sgA (purple), is found in the cell's genome, but, undesired cutting by other sgRNAs may have occurred, illustrated by the sgB (orange). This is tolerable due to the weaker promoter for sgRNAs (U6:2). (B) In the v.2 approach, anti-CRISPR (red) is expressed, initially inactivating Cas9 and preventing CRISPR, while one sgRNA is still integrated into the cell's genome using ϕ C31 integration. Later, following cell divisions, transiently transfected sgRNAs are lost and the integrated sgRNA alone is active in the cell. We also used the stronger U6:3 promoter in v.2 to express more sgRNA. pInt = ϕ C31 Integrase expression plasmid. pIntAC = ϕ C31-T2A-AcrIIa4 expression plasmid. (C) T7 endonuclease I-sensitivity assay, showing the edited versus unedited allele targeted by a CRISPR sgRNA measured over time in cells transfected as indicated. Cell populations transfected with IntAC exhibited a clear early delay in editing, with editing efficiency returning to nearly that of controls by day 18. Quantification of editing (edited band intensity divided by the sum of the intensities of the edited and unedited bands). (D) Phenotypic validation of IntAC-delayed editing. Cells with Rho1 suppression, characterized by isotropically overgrown cell morphology, were observed later in the IntAC population compared to controls, confirming the temporal delay in Cas9 activity. Red = mCherry in *Drosophila* S2R+ derivative PT5 (NPT005; DGRC #229) cells; Green = Free GFP expression from sgRNA plasmid (pLib6.6/sgRho1) which additionally encodes Actin promoter-driven GFP.

8. *Figures 5 a and c are quite confusing and insufficiently labeled and explained. For example, what are the green, yellow, purple and orange boxes, triangles and ovals?*

We now break up Figure 5 into two figures, allowing us more space to include a legend for the various modifications within the GPI anchor synthesis pathway, now added to panel A.

9. *Results page 5. "Moreover, this increase was not observed when the anti-CRISPR protein was left out (v.2 library with ϕ C31 integrase alone, 'Int') (Figure 3A)." Fig.3A also nicely shows how sgRNA targeting non-expressed genes are associated with strong effects in this setup. You make this point in the figure legend, but I would also mention it in the main text, as it is important and illustrates why inhibition is so important.*

We thank the reviewer for highlighting this important point. We have now changed the language to emphasize this.

We conducted genome-wide fitness screens using the v.2 screening library with or without IntAC, and compared the results to v.1 screen data [2] and genome-wide arrayed RNAi screens [29]. For each method, we binned genes based on their predicted likelihood of being fitness genes (from low to high MLE Z-score). Because genes required for cell fitness must also be expressed as RNA, we looked at the RNA

expression of genes in each bin. As expected, genes in lower bins were highly enriched for expressed genes and had very few non-expressed genes, unlike intermediate or high bins. The strength of this enrichment indicates how many fitness genes were uncovered by each method and thus serves as an indicator of screen quality. Compared to v.1, the v.2 screen with IntAC had more bins enriched in expressed genes, indicating a higher number of fitness genes detected and improved screen quality (Figure 3A). Furthermore, running the v.2 screen without anti-CRISPR led to a marked decrease in fitness genes detected and therefore lower screen quality (Figure 3A). This finding illustrates why anti-CRISPR inhibition is crucial when using the v.2 library and strongly supports our hypothesis (and pilot data) suggesting that early, unregulated Cas9 activity in the absence of anti-CRISPR diminishes screen precision, especially when paired with elevated sgRNA levels driven by the stronger dU6:3 promoter.

Based on another reviewer point, we also changed Figure 3A to make it clearer to the reader that fitness gene assignment is dramatically worsened by leaving out anti-CRISPR.

10. *Page 5: “These results suggest that much of the improvement between v.1 and v.2 is driven by higher sgRNA expression levels via the dU6:3 promoter as made possible by use of the IntAC approach”. I guess another important factor is that there is a better 1-to-1 relationship between detected sgRNA and edited gene(s) in v.2.*

That was, indeed, our meaning. We changed the language to make this point more clearly.

These results suggest that much of the improvement between v.1 and v.2 is driven by higher sgRNA expression levels via the dU6:3 promoter and the improved 1-to-1 relationship between detected sgRNA and edited genes enabled by the use of anti-CRISPR. We hypothesize that sgRNA design optimization using machine learning had a relatively minor impact.

11. *Introduction: “Tighter strategies for drug-controlled expression such as ‘tet-on’ are lacking in Drosophila.” - Tet-on is also notoriously leaky in mammalian cells when combined with Cas9, so you might want to rephrase or remove that sentence.*

We thank the reviewer for their experience with Tet-on, which we have not used, ourselves. Taking this advice, we have removed the reference to tet-on.

12. *Typo in Figure 2b: targeting*

Corrected

13. *Suppl. Fig. 2a: Indicate on the y axis that these are consecutive weeks.*

Added

Reviewer 2

1. *“Fitness genes” might not mean much to many particularly in a cell-specific invitro context, rather than actual living insects, which should probably be specified or changed to “essential” for Drosophila cells?*

We use ‘fitness’ rather than ‘essential’ because these screens are unable to distinguish genes whose perturbation would result in death versus slow growth. The statistics of CRISPR screens are such that all sgRNAs for a given gene resulting in slow growth might be a stronger signal than half the sgRNAs resulting in cell death. A pioneering study in this field, Hart et al., 2015 (Cell), innovated this term. We now define ‘fitness genes’ in our introduction and cite Hart et al.

Fitness genes are defined as those genes needed for optimal cell proliferation [21].

2. *“AcrIIa4 is the most potent known protein inhibitor of Cas9 activity [17-19]. This protein mimics guide RNAs, binding and functionally inactivating Cas9. Cas9 activity is thereby...” My understanding is that Acr proteins mimic DNA binding rather than gRNA, which I believe is still able to form the RNP complex. The latter is unable to interact with the DNA target due to the presence of the Acr. Please doublecheck and eventually revise accordingly.*

We thank the author for bringing this misunderstanding to our attention. We changed the text to:

This protein binds to the Cas9–sgRNA complex mimicking DNA, obstructing the PAM recognition site, and impeding the necessary conformational changes.

3. *“This finding supports our hypothesis and pilot experiments suggesting that early, uncontrolled Cas9 activity in the absence of anti-CRISPR reduces the precision of the screen, particularly when combined with elevated sgRNA level due to our use of the stronger dU6:3 promoter.” Is this assumed from the wider positive z-score shown in 3A top right plot? It might be useful to expand a little more to better explain this.*

We thank the author for bringing this important misunderstanding to our attention. We changed the text to:

This protein binds to the Cas9–sgRNA complex mimicking DNA, obstructing the PAM recognition site, and impeding the necessary conformational changes.

4. *“This finding supports our hypothesis and pilot experiments suggesting that early, uncontrolled Cas9 activity in the absence of anti-CRISPR reduces the precision of the screen, particularly when combined with elevated sgRNA level due to our use of the stronger dU6:3 promoter.” Is this assumed from the wider positive z-score shown in 3A top right plot? It might be useful to expand a little more to better explain this.*

To address this confusion in the “Z-score vs. expression plots” we now bin the genes by their predicted importance for fitness using the various approaches (genome-wide RNAI,

CRISPR screen v.1, CRISPR screen v.2 with anti-CRISPR, CRISPR screen v.2 without anti-CRISPR) and display the distribution of expression levels for genes within each bin. We also explain this better in the text:

We conducted genome-wide fitness screens using the v.2 screening library with or without IntAC, and compared the results to v.1 screen data [2] and genome-wide arrayed RNAi screens [29]. For each method, we binned genes based on their predicted likelihood of being fitness genes (from low to high MLE Z-score). Because genes required for cell fitness must also be expressed as RNA, we looked at the RNA expression of genes in each bin. As expected, genes in lower bins were highly enriched for expressed genes and had very few non-expressed genes, unlike intermediate bins. The strength of this enrichment indicates how many fitness genes were uncovered by each method and thus serves as an indicator of screen quality. Compared to v.1, the v.2 screen with IntAC had more bins enriched in expressed genes, indicating a higher number of fitness genes detected and improved screen quality (Figure 3A). Furthermore, running the v.2 screen without anti-CRISPR led to a marked decrease in fitness genes detected and therefore lower screen quality (Figure 3A). This finding illustrates why anti-CRISPR inhibition is crucial when using the v.2 library and strongly supports our hypothesis (and pilot data) suggesting that early, unregulated Cas9 activity in the absence of anti-CRISPR diminishes screen precision, especially when paired with elevated sgRNA levels driven by the stronger dU6:3 promoter.

5. *“This work underscores the utility of straightforward single-gene knockout screens in Drosophila cells to identify genes that would be missed in similar screens in mammalian cells.” I assume this would only be applicable to conserved genes/phenotypes. I would suggest specifying the caveat.*

We now make this important point.

“For conserved gene families, this work underscores the utility of straightforward single-gene knockout screens in *Drosophila* cells to identify genes that would be missed in similar screens in mammalian cells.”

6. *Figure 1A/B. I can't see a description of the plasmid indicated as “plnt” in the figure Figure 4B and related text “1 to many” is not very helpful to indicate if a specific N of orthologues threshold is applied or any gene >1 orthologues is considered as “many”. Please change/specify accordingly*

We now add descriptions of pInt and pIntAC to the Figure 1 legend.

Regarding Figure 4, the DIOPT method uses protein sequence to score ortholog relationships, and the cutoff was made using this score alone; not from the number of orthologs. To clarify these important points, we make a changes to the Figure 4 legend as follows:

Figure 4: Correlation between Drosophila fitness genes and human orthologs in IntAC screens. (A) For genes with a 1-to-1 ortholog relationship between Drosophila and humans, a strong correlation ($R^2 = 0.41$) was observed, suggesting high conservation of fitness genes across species. Schematic of processing steps comparing Drosophila orthologs to their predicted human orthologs (assessed by a DIOPT score >6 , using the DRSC Integrative Ortholog Prediction Tool [32] **which integrates multiple methods of assessing gene orthology from gene sequence**). Genes with a 1-to-1 relationship were retained for analysis, and the first-quartile (Q1) of their Cas9-mediated Essentiality Ranking Evaluator of Specificity (CERES) scores was plotted across all DepMap data (23Q4) [33]. (B) **Any gene found to have more than 1 ortholog at a DIOPT score of greater than 6 was considered to have a 1-to-Many relationship (Drosophila genes with multiple human orthologs)**. In these cases, the correlation with fitness scores dropped significantly ($R^2 = 0.18$). Schematic showing processing steps. When multiple human orthologs had a DIOPT score > 6 , the ortholog with the lowest CERES score was chosen in order to counter-bias the data against selecting an inactive paralog. This result suggests that paralog redundancy in humans can mask the essentiality of certain genes.

Reviewer 3

1. (1) *Do the authors have any data to confirm that only one gRNA becomes integrated into each attP site/cell? This would be important to properly interpret the specificity and sensitivity data presented later in the paper.*

AttP sites were delivered using a transposon that mobilizes with low frequency followed by single-cell cloning. We have not confirmed the number of insertions beyond the initial study (Neumüller et al., 2012 Genetics) in which inverse PCR was used to detect transposon insertions, and only one insertion was found. We believe the comparison between the v.1 and v.2 methods are valid in any case because the same clonal cell population was used for both screens.

2. (2) *In Fig 1, the authors do not show that editing in their new system reaches a maximum 18 days post-transfection, as this is the latest time point tested. They would need to test additional (later) time points to make this claim.*

We agree with the reviewer that wording is imprecise. When we quantify the edited versus unedited bands using T7 endonuclease I assays, we observe near convergence around Day 18 (76.8% vs. 65.5%), which is why we stopped measuring at this point, but it is true that we do not have enough information to conclude that this is the maximum editing efficiency. We reworded the reference to Figure 1C as follows:

Editing reached a maximum (of ~75%) 10-14 days after transfection when the sgRNA vector was transfected along with Int. However, the presence of the IntAC plasmid imparted a clear delay; for the IntAC population, editing efficiency was initially low on days 10 and 14 but then reached nearly the level of the Int population (~65%) on day 18 (Figure 1C).

To make these details easier to follow, we also now provide precise quantification of editing efficiency underneath the agarose gel in the figure.

3. *(3) The authors state in the Results section that their AI-improved gRNA design had only a minor impact on the improvements to their screen. However, as they acknowledge in the Discussion, they do not actually have any evidence to support this claim. An additional screen, potentially with a subset of genes, would be required to support this claim.*

We thank the reviewer for noticing that it will be important to determine the contribution of AI-driven guide improvement. The most efficient way to test this is to further iterate on top of this U6:3-driven platform using machine learning. This is the focus of our future studies.

4. *(4) In Fig 5, the authors note that several genes essential for GPI anchor protein synthesis and complex N-glycosylation pathways were not picked up in their PA resistance screen. To more firmly verify the specificity of their improved screen, the authors would need to transiently silence (or knock out) the three genes that have not previously been investigated for a role in these pathways.*

We agree that exploring the contribution of the five genes not retrieved in the screen is important, but it lies beyond the scope of this study. Our objective here was to identify the major GPI anchor synthesis genes necessary for PA resistance, rather than to verify every predicted hit. Moreover, we now apply the “homology-directed insertion” approach to successfully generate a PIGA mutant that displays a lack of surface GPI anchors (Supplementary Figure 5C), which illustrates an approach that can be extended to individually knock out these five genes in future work. Thus, we believe that detailed functional studies of these candidates are outside the aims of this manuscript.

5. *(5) CG46311 is identified as a previously unidentified orthologue of human PIG-Y (previously thought not to have an orthologue in Drosophila). These findings would be more convincing if sequence alignments and structural similarity comparisons of the Alphafold structural predictions of PIG-Y and CG46311 were presented.*

Spurred by the reviewer, we undertook Alphafold modeling of the entire 7-member GnT complex. Remarkably, we see CG46311 binding tightly to multiple components of the GnT complex (by Alphafold-Multimer scoring metrics) and adopting a similar position in the complex compared with PIG-Y’s position in the human GnT complex. We have now added this data to Figure 6 and Supplementary Figure 5.

6. *(6) The microscopy images in Fig 1F are difficult to interpret. Confocal images and quantification should be shown to strengthen these data as they are essential for confirming the function of the putative novel PIG-Y orthologue.*

We believe the reviewer is referring to Figure 5F. We have now split Figure 5 into two

figures. We improved our live cell confocal imaging and moved this to Supplementary Figure 5. In Figure 6, we now use fixed but not permeabilized cell staining and confocal imaging to illustrate the failure of CG46311 cells to display GFP on the cell surface, and the rescue of this defect with CG46311 cDNA.

7. (1) A number of terms used may not be clear to the broad audience for which this manuscript will be of interest. Please define “isotropically overgrown”, “dropout” and “passengers” to improve the clarity of the manuscript.

We removed “isotropically overgrown” and now define ‘dropout’ and ‘passenger’ along with their first use in the text.

8. (2) Can the authors explain how *Rho1* appears in their list of essential genes when they were able to target this gene in their validation experiments in Fig 1D?

We thank the reviewer for pointing this out. *Rho1* knockout causes cells to become large and multinucleate initially, but after several weeks, they begin to die by an unknown process. We have added this information to the text.

Rho1 knockout cells initially become large [23] due to a failure in cytokinesis and die by an unknown process after several weeks (data not shown).

9. (3) In Fig 1, a key or more clear description in the figure legend would be helpful to make Fig 1A and B easier to interpret.

We now add to the Figure 1 legend:

In the v1 approach, sgRNAs are active upon pooled transfection and one of these at random integrates into the cell’s genome via ϕ C31 integration. Following cell divisions and the loss of transiently transfected plasmids, one sgRNA, illustrated by *sgA* (purple), is found in the cell’s genome, but, undesired cutting by other sgRNAs may have occurred, illustrated by the *sgB* (orange). This is tolerable due to the weaker promoter for sgRNAs (*U6:2*). (B) In the v.2 approach, anti-CRISPR (red) is expressed, initially inactivating Cas9 and preventing CRISPR, while one sgRNA is still integrated into the cell’s genome using ϕ C31 integration. Later, following cell divisions, transiently transfected sgRNAs are lost and the integrated sgRNA alone is active in the cell. We also used the stronger *U6:3* promoter in v.2 to express more sgRNA. pInt = ϕ C31 Integrase expression plasmid. pIntAC = ϕ C31-T2A-AcrIIa4 expression plasmid.

10. (4) In Fig 1D, what do the green/red colours represent in the microscopy images?

We add to the Figure 1 legend:

**Red = mCherry in *Drosophila* S2R+ derivative PT5 (NPT005; DGRC #229) cells;
Green = Free GFP expression from sgRNA plasmid (*pLib6.6/sgRho1*) which**

additionally encodes *Actin* promoter-driven GFP.

11. (5) I found the inclusion of the Alphafold structural predictions in Fig 5F distracting and unnecessary.

We have significantly reworked the figure and the Alphafold modeling component and hope the reviewer will find it more acceptable.